# Smooth Gradients, Stable Learning: Logits Convexity for Reinforcement Learning

## Abstract

Reinforcement learning (RL) has been pivotal to the recent success of large language models (LLMs) across a broad spectrum of tasks. However, RL optimization often suffers from inherent stability challenges, particularly when compared to supervised fine-tuning (SFT). In this work, we investigate the stability gap between SFT and RL from a gradient-based perspective. We identify a property of the cross-entropy loss with softmax in SFT, which we term *logits convexity*, characterized by local convexity with respect to logits. Our theoretical analysis shows that logits convexity induces smoother gradient magnitudes during optimization, thereby enhancing stability. In contrast, the policy gradient objectives of widely used algorithms such as PPO and GRPO lack this property. Motivated by this insight, we propose Logits Convex Optimization (LCO), a simple yet effective policy optimization strategy to align the policy distribution with a carefully designed target distribution via KL divergence to emulate the stabilizing effects of logits convexity. Empirical results demonstrate that LCO improves stability and consistently outperforms conventional RL methods on both reasoning and non-reasoning benchmarks. Code and datasets will be made publicly available.

## 1 Introduction

Reinforcement learning (RL) has become a cornerstone for aligning large language models (LLMs) with human preferences (Ouyang et al., 2022; Bai et al., 2024) and enhancing complex capabilities such as reasoning (Guo et al., 2025; Yang et al., 2025a). Despite these advances, RL training often suffers from inherent instability (Rafailov et al., 2024). Existing approaches attempt to address this issue through variance reduction in advantage estimation (Schulman et al., 2015b), clipping strategies that constrain parameter updates (Schulman et al., 2017; Yu et al., 2025), and KL-based penalties that regulate policy shifts (Ouyang et al., 2022; Shao et al., 2024). Although these solutions mitigate instability to some extent, they do not fully resolve it (Team et al., 2025; Zhu et al., 2025a). This motivates a deeper understanding of the underlying causes of RL instability in LLMs.

In this work, we analyze RL instability from a gradient-based perspective. We observe that the loss functions in widely used RL algorithms, such as PPO (Schulman et al., 2017), often exhibit large fluctuations or explosions in gradient magnitude as training progresses (Figure 1(a)). These fluctuations can induce excessive parameter updates, potentially leading to training collapse (Figure 1(b)). By contrast, supervised fine-tuning (SFT) typically demonstrates more stable optimization throughout training (Wu et al., 2025; He et al., 2025; Liu et al., 2025). This observation naturally raises the question: *what accounts for the greater stability of SFT compared to RL methods?*

Upon examining the underlying causes, we identify a property termed *logits convexity*, defined as local convexity at the logits level. Our theoretical analysis demonstrates that logits convexity facilitates favorable gradient behavior during optimization, naturally leading to diminishing gradient magnitudes as the policy approaches convergence. This behavior aligns with the intuitive expectation that updates should become more conservative near an

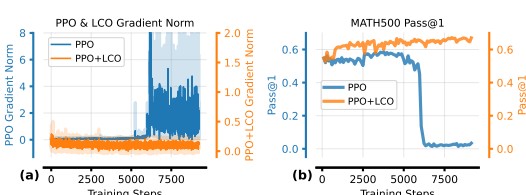

Figure 1: **(a)** Gradient norm during training for PPO and PPO+LCO. **(b)** Pass@1 results of PPO and PPO+LCO on the MATH500 benchmark.

optimum. While SFT loss exhibits logits convexity, which ensures stable gradient updates, RL objectives such as REINFORCE (Williams, 1992) and PPO (Schulman et al., 2017) lack this property, making them susceptible to large gradient fluctuations and training instability.

Building on this property, we propose **Logits Convex Optimization (LCO)**, an RL optimization objective that preserves logits convexity and promotes stable training. LCO works by aligning the policy distribution with a carefully designed target distribution through KL divergence. This target distribution preserves the core objective of policy gradient methods: it encourages the policy to increase the probability of beneficial actions while suppressing the probability of undesirable actions. LCO can be seamlessly incorporated into existing RL algorithms like PPO. With this integration, LCO produces stable gradient updates (Figure 1(a)) and delivers consistent performance improvements (Figure 1(b)). Empirical evaluations on both reasoning and non-reasoning tasks show that LCO achieves superior stability and performance compared to standard RL baselines. Furthermore, our analysis yields three key findings. **First**, we identify a primary source of training instability in standard RL: excessively large gradient norms arising from negative samples in non-convex loss regions. **Second**, we reveal that sampled actions with low probability can cause sudden spikes in gradient updates, which affect the stability of methods such as PPO and GRPO. **Third**, we show that preserving logit convexity during optimization leads to stable and diminishing gradient updates as training approaches convergence, which mitigates RL training instability.

## 2 PRELIMINARY

### 2.1 NOTATION AND SUPERVISED FINE-TUNING

We define the state $s_t$ at time step $t$ as the combination of the prompt tokens and all tokens generated up to that step. An action $a_{t,i}$ at time step $t$ corresponds to selecting the $i$-th token from the vocabulary $\mathcal{A}$. Given state $s_t$, the probability that the policy $\pi_\theta$ generates action $a_{t,i}$ is denoted by $\pi_\theta(a_{t,i}|s_t)$. In this work, we consider the policy $\pi_\theta$ to be a language model with a softmax output:

$$\pi_\theta(a_{t,i}|s_t) = \frac{\exp z_\theta(a_{t,i}|s_t)}{\sum_k \exp z_\theta(a_{t,k}|s_t)}, \tag{1}$$

where $z_\theta(a_{t,i}|s_t)$ is the logit corresponding to the $i$-th action at time step $t$, parameterized by $\theta$. In the following, we use $i$ to denote the index of a sampled action $a_{t,i}$, $j$ the index of a non-sampled action $a_{t,j}$, and $k$ the index of an arbitrary action $a_{t,k}$.

Supervised fine-tuning (SFT) trains language models to maximize the likelihood of target tokens given input text. Given context $s_t$ and target token $a_{t,i}$ at time step $t$, the loss function is defined as:

$$\mathcal{L}_{\text{SFT}}^t = -\log \pi_\theta(a_{t,i}|s_t). \tag{2}$$

### 2.2 POLICY GRADIENT

Policy gradient (PG) methods are a class of RL algorithms that optimize policy $\pi_\theta$ by estimating the gradient of the expected return. At time step $t$, the standard PG loss function is defined as:

$$\mathcal{L}_{\text{PG}}^t = -\Psi_{t,i} \log \pi_\theta(a_{t,i}|s_t), \tag{3}$$

where $\Psi_{t,i}$ represents either the return or the advantage for sampled action $a_{t,i}$ at time step $t$. REINFORCE (Williams, 1992) is a canonical example of a PG method.

### 2.3 POLICY GRADIENT WITH IMPORTANCE SAMPLING

Policy gradient with importance sampling (denoted PG-IS) methods mitigate the sample inefficiency inherent in standard PG methods. By introducing importance sampling weights, these methods allow policy updates to reuse samples generated by an older policy $\pi_{\theta_{\text{old}}}$, rather than relying on samples from the current policy. At time step $t$, the loss function is defined as:

$$\mathcal{L}_{\text{PG-IS}}^t = -\Psi_{t,i} \frac{\pi_\theta(a_{t,i}|s_t)}{\pi_{\theta_{\text{old}}}(a_{t,i}|s_t)}. \tag{4}$$

A representative PG-IS method is proximal policy optimization (PPO) (Schulman et al., 2017).

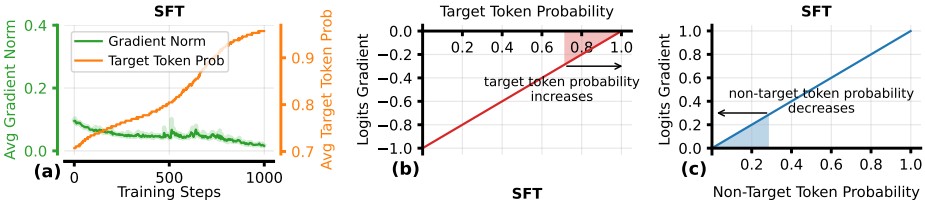

Figure 2: Training dynamics of supervised fine-tuning (SFT). **(a)** Average (Avg) gradient norm, $\|\nabla_\theta \mathcal{L}_{\text{SFT}}^t\|$, decreases as training progresses while average target token probability (prob) on training samples increases. **(b)** Target token logit gradient: $\partial \mathcal{L}_{\text{SFT}}^t / \partial z_\theta(a_{t,i}|s_t) = \pi_\theta(a_{t,i}|s_t) - 1$. As $\pi_\theta(a_{t,i}|s_t) \to 1$, this gradient approaches 0. **(c)** Non-target token logit gradient: $\partial \mathcal{L}_{\text{SFT}}^t / \partial z_\theta(a_{t,j}|s_t) = \pi_\theta(a_{t,j}|s_t)$. As $\pi_\theta(a_{t,j}|s_t) \to 0$, this gradient also approaches 0.

## 3 GRADIENT DYNAMICS

In this section, we empirically analyze the different gradient dynamics of $\mathcal{L}_{\text{SFT}}^t$, $\mathcal{L}_{\text{PG}}^t$, and $\mathcal{L}_{\text{PG-IS}}^t$. We then demonstrate how gradient dynamics relate to logits gradients and affect training stability.

### 3.1 GRADIENT DYNAMICS OF SFT

We first provide the gradient of SFT loss $\mathcal{L}_{\text{SFT}}^t$ with respect to parameters $\theta$:

$$\nabla_\theta \mathcal{L}_{\text{SFT}}^t = \sum_k^{|\mathcal{A}|} \left[ \frac{\partial \mathcal{L}_{\text{SFT}}^t}{\partial z_\theta(a_{t,k}|s_t)} \nabla_\theta z_\theta(a_{t,k}|s_t) \right], \tag{5}$$

where $|\mathcal{A}|$ is the size of the vocabulary. For a logit $z_\theta(a_{t,k}|s_t)$, the gradient of $\mathcal{L}_{\text{SFT}}^t$ with respect to $z_\theta(a_{t,k}|s_t)$ is given by (refer to Appendix F.1 for a detailed derivation):

$$\frac{\partial \mathcal{L}_{\text{SFT}}^t}{\partial z_\theta(a_{t,k}|s_t)} = \pi_\theta(a_{t,k}|s_t) - \delta_{ik}, \tag{6}$$

where $i$ denotes the index of the target token $a_{t,i}$, $k$ denotes the index of an arbitrary token $a_{t,k}$ in the vocabulary, and $\delta_{ik}$ is the Kronecker delta, defined as $\delta_{ik} = 1$ if $i = k$ and $\delta_{ik} = 0$ otherwise.

The logit gradient for a target token $a_{t,i}$ is $\frac{\partial \mathcal{L}_{\text{SFT}}^t}{\partial z_\theta(a_{t,i}|s_t)} = \pi_\theta(a_{t,i}|s_t) - 1$, whereas for a non-target token $a_{t,j}$ ($j \neq i$), it is $\frac{\partial \mathcal{L}_{\text{SFT}}^t}{\partial z_\theta(a_{t,j}|s_t)} = \pi_\theta(a_{t,j}|s_t)$. Figure 2(a) illustrates the overall gradient dynamics, while Figures 2(b) and (c) depict the logit gradient dynamics. During training, target token probabilities increase and gradient norms decrease. A similar trend is observed in the logit gradients: as target token probabilities approach 1 and non-target probabilities approach 0, the corresponding logit gradient magnitudes diminish, reflecting convergence. This behavior aligns with the intuition that as model nears optimality and loss decreases, the parameter updates naturally become smaller.

### 3.2 GRADIENT DYNAMICS OF POLICY GRADIENT

For a logit $z_\theta(a_{t,k}|s_t)$, the gradient of $\mathcal{L}_{\text{PG}}^t$ with respect to $z_\theta(a_{t,k}|s_t)$ is given by:

$$\frac{\partial \mathcal{L}_{\text{PG}}^t}{\partial z_\theta(a_{t,k}|s_t)} = \Psi_{t,i}(\pi_\theta(a_{t,k}|s_t) - \delta_{ik}). \tag{7}$$

The detailed derivation is provided in Appendix F.2. Since the scalar $\Psi_{t,i}$ only scales the gradients without changing their direction, its magnitude does not affect our analysis. Therefore, for $\Psi_{t,i} > 0$, we set $\Psi_{t,i} = 1$ for simplicity. Under this setting, Equation 7 reduces to Equation 6. In other words, when $\Psi_{t,i}$ is positive, $\mathcal{L}_{\text{PG}}^t$ exhibits gradient dynamics analogous to those of $\mathcal{L}_{\text{SFT}}^t$: as training progresses, the probability of the sampled action increases while that of non-sampled actions decreases, leading to a reduction in the overall gradient norm. When $\Psi_{t,i} < 0$, we set $\Psi_{t,i} = -1$ for simplicity. In this case, the logit gradient of the sampled action $a_{t,i}$ becomes $\frac{\partial \mathcal{L}_{\text{PG}}^t}{\partial z_\theta(a_{t,i}|s_t)} = 1 - \pi_\theta(a_{t,i}|s_t)$, while for any non-sampled action $a_{t,j}$ ($j \neq i$) it becomes $\frac{\partial \mathcal{L}_{\text{PG}}^t}{\partial z_\theta(a_{t,j}|s_t)} = -\pi_\theta(a_{t,j}|s_t)$. We visualize the overall gradient dynamics in Figure 3(a), and the logit gradient dynamics in Figures 3(b) and (c). **A counterintuitive phenomenon emerges: as training progresses, the loss decreases while the gradient norms grow.** Likewise, as the probability of the sampled action decreases and the probability of the non-sampled action increases, the magnitude of the logit gradients increases.

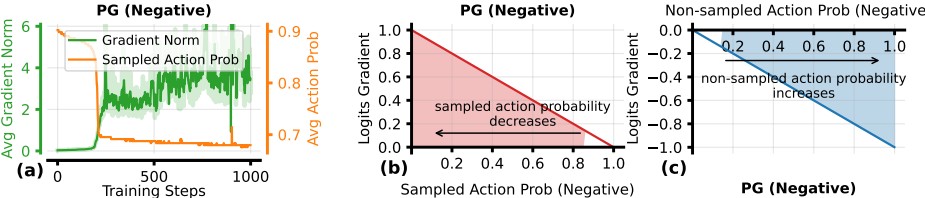

Figure 3: Policy gradient (PG) training dynamics on negative actions ($\Psi_{t,i} < 0$). **(a)** For negative actions, gradient norm $\|\nabla_\theta \mathcal{L}_{\mathrm{PG}}^t\|$ oscillates as training progresses while sampled action probabilities decrease. **(b)** Sampled action logit gradient: $\partial \mathcal{L}_{\mathrm{PG}}^t / \partial z_\theta(a_{t,i}|s_t) = 1 - \pi_\theta(a_{t,i}|s_t)$. As $\pi_\theta(a_{t,i}|s_t) \to 0$, this gradient magnitude increases. **(c)** Non-sampled action logit gradient: $\partial \mathcal{L}_{\mathrm{PG}}^t / \partial z_\theta(a_{t,j}|s_t) = -\pi_\theta(a_{t,j}|s_t)$. As $\pi_\theta(a_{t,j}|s_t) \to 1$, this gradient magnitude also increases.

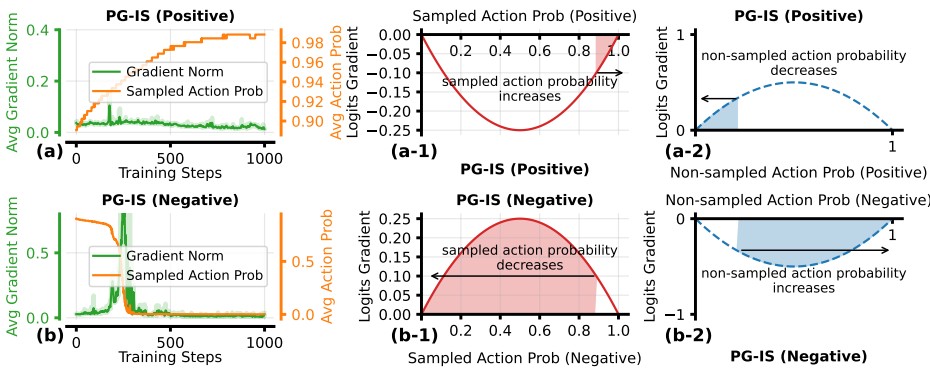

Figure 4: Training dynamics of policy gradient with importance sampling (PG-IS). **(a)** For positive actions ($\Psi_{t,i} > 0$), gradient norm $\|\nabla_\theta \mathcal{L}_{\mathrm{PG-IS}}^t\|$ decreases as training progresses while sampled action probabilities increase. **(a-1)** Sampled action logit gradient: $\partial \mathcal{L}_{\mathrm{PG-IS}}^t / \partial z_\theta(a_{t,i}|s_t) = \pi_\theta(a_{t,i}|s_t)(\pi_\theta(a_{t,i}|s_t) - 1)$. As $\pi_\theta(a_{t,i}|s_t) \to 1$, this gradient approaches 0. **(a-2)** Non-sampled action logit gradient: $\partial \mathcal{L}_{\mathrm{PG-IS}}^t / \partial z_\theta(a_{t,j}|s_t) = \pi_\theta(a_{t,i}|s_t)\pi_\theta(a_{t,j}|s_t)$. As $\pi_\theta(a_{t,j}|s_t) \to 0$, this gradient approaches 0. **(b)** For negative actions ($\Psi_{t,i} < 0$), gradient norm exhibits an initial increase followed by a decrease as training progresses, while sampled action probabilities decrease. **(b-1)** Sampled action logit gradient: $\partial \mathcal{L}_{\mathrm{PG-IS}}^t / \partial z_\theta(a_{t,i}|s_t) = \pi_\theta(a_{t,i}|s_t)(1 - \pi_\theta(a_{t,i}|s_t))$. As $\pi_\theta(a_{t,i}|s_t) \to 0$, this gradient magnitude exhibits an initial increase followed by a decrease. **(b-2)** Non-sampled action logit gradient: $\partial \mathcal{L}_{\mathrm{PG-IS}}^t / \partial z_\theta(a_{t,j}|s_t) = -\pi_\theta(a_{t,i}|s_t)\pi_\theta(a_{t,j}|s_t)$. As $\pi_\theta(a_{t,j}|s_t) \to 1$, the gradient value initially decreases before gradually increasing, while its magnitude exhibits the opposite trend.

### 3.3 GRADIENT DYNAMICS OF POLICY GRADIENT WITH IMPORTANCE SAMPLING

For policy gradient with importance sampling (PG-IS), the gradient of $\mathcal{L}_{\mathrm{PG-IS}}^t$ with respect to the logit $z_\theta(a_{t,k}|s_t)$ is given by (derivation in Appendix F.3):

$$\frac{\partial \mathcal{L}_{\mathrm{PG-IS}}^t}{\partial z_\theta(a_{t,k}|s_t)} = \frac{\Psi_{t,i}}{\pi_{\theta_{\mathrm{old}}}(a_{t,i}|s_t)} \pi_\theta(a_{t,i}|s_t)(\pi_\theta(a_{t,k}|s_t) - \delta_{ik}) \tag{8}$$

For simplicity, we absorb $\pi_{\theta_{\mathrm{old}}}(a_{t,i}|s_t)$ into $\Psi_{t,i}$ and analyze two cases: (1) For $\Psi_{t,i} > 0$, we set $\Psi_{t,i} = 1$. Then the logits gradient of sampled action is $\frac{\partial \mathcal{L}_{\mathrm{PG-IS}}^t}{\partial z_\theta(a_{t,i}|s_t)} = \pi_\theta(a_{t,i}|s_t)(\pi_\theta(a_{t,i}|s_t) - 1)$, while for a non-sampled action, it is $\frac{\partial \mathcal{L}_{\mathrm{PG-IS}}^t}{\partial z_\theta(a_{t,j}|s_t)} = \pi_\theta(a_{t,i}|s_t)\pi_\theta(a_{t,j}|s_t)$. As shown in Figure 4(a), gradient norm decrease as training progress, while magnitude of logit gradients decrease (Figures 4(a-1) and (a-2)). (2) For $\Psi_{t,i} < 0$, we set $\Psi_{t,i} = -1$. In this case, the gradient dynamics behave differently. Figure 4(b) shows that gradient norm of $\mathcal{L}_{\mathrm{PG-IS}}^t$ exhibits initial increase followed by decrease as training progresses. A similar phenomenon can also be observed in logit gradients (Figure 4(b-1) and (b-2)). Gradient magnitude spikes typically occur for sampled actions with low probabilities (near 0.5), causing large parameter updates that can destabilize training.

## 4 ON THE CONVEXITY OF LOGITS

Previous analysis shows that, compared to $\mathcal{L}_{\mathrm{SFT}}^t$, $\mathcal{L}_{\mathrm{PG}}^t$ and $\mathcal{L}_{\mathrm{PG-IS}}^t$ are more susceptible to unstable training. In this section, we conduct a deeper investigation and identify an important property: the *convexity* exhibited at the logits level plays a critical role in ensuring smooth and stable convergence.

## 4.1 DEFINITION OF LOGITS CONVEXITY

**Definition 1** (Logits Convexity). *Let $\mathcal{L} : \mathbb{R}^n \to \mathbb{R}$ be a twice-differentiable loss function that takes logits $\boldsymbol{z}_\theta \in \mathbb{R}^n$ parameterized by $\theta$ as input. We say that $\mathcal{L}$ is logits convex if and only if the Hessian matrix of $\mathcal{L}$ with respect to $\boldsymbol{z}_\theta$ is positive semi-definite.*

To further illustrate the property of logits convexity, we first present two fundamental propositions.

**Proposition 1.** *Let $\mathcal{L} : \mathbb{R}^n \to \mathbb{R}$ be a twice-differentiable loss function taking logits $\boldsymbol{z}_\theta \in \mathbb{R}^n$ parameterized by $\theta$ as input. Let $\boldsymbol{z}_\theta^* \in \mathbb{R}^n$ be the optimal logits. If $\mathcal{L}$ is logits convex, then:*

$$\lim_{\boldsymbol{z}_\theta \to \boldsymbol{z}_\theta^*} \|\nabla_\theta \mathcal{L}\| = 0. \tag{9}$$

*Proof.* See Appendix G. □

Proposition 1 highlights a key property of logits convexity: as the logits approach their optimal values, the gradient converges to zero, which can help prevent gradient divergence during training.

**Proposition 2.** *Let $\mathcal{L} : \mathbb{R}^n \to \mathbb{R}$ be a twice-differentiable loss function that takes logits $\boldsymbol{z}_\theta \in \mathbb{R}^n$ parameterized by $\theta$ as input. Let $z_{\theta,i}$ denote the $i$-th element of $\boldsymbol{z}_\theta$. Let $z'_{\theta,i}$ and $z''_{\theta,i}$ be two values on the same side of the optimal value $z^*_{\theta,i}$, with $z''_{\theta,i}$ closer to $z^*_{\theta,i}$ than $z'_{\theta,i}$:*

$$\left| z''_{\theta,i} - z^*_{\theta,i} \right| < \left| z'_{\theta,i} - z^*_{\theta,i} \right|. \tag{10}$$

*If $\mathcal{L}$ is logits convex, then the logit gradient magnitudes satisfy the following relationship:*

$$\left| \frac{\partial \mathcal{L}}{\partial z''_{\theta,i}} \right| \le \left| \frac{\partial \mathcal{L}}{\partial z'_{\theta,i}} \right|. \tag{11}$$

*Proof.* See Appendix H. □

Proposition 2 shows that the logit gradient magnitude decreases monotonically as logits approach their optimal values. Since the parameter gradient norm can be written as:

$$\left\| \frac{\partial \mathcal{L}}{\partial z_{\theta,i}} \nabla_\theta z_{\theta,i} \right\| = \underbrace{\left| \frac{\partial \mathcal{L}}{\partial z_{\theta,i}} \right|}_{\text{scaling factor}} \left\| \nabla_\theta z_{\theta,i} \right\|, \tag{12}$$

the logit gradient serves as a global scaling factor that modulates the magnitude of parameter updates. Consequently, logits convexity ensures that parameter gradients decrease smoothly during optimization, thereby reducing the risk of sudden gradient spikes or unstable updates.

Below, we present a series of propositions to analyze the logits convexity of different loss functions.

**Proposition 3.** *The supervised fine-tuning loss function $\mathcal{L}^t_{\text{SFT}}$, as defined in Equation 2, is logits convex at each time step (Proof. See Appendix I.1).*

**Proposition 4.** *The policy gradient loss function $\mathcal{L}^t_{\text{PG}}$, as defined in Equation 3, is logits convex at time steps where $\Psi_{t,i} > 0$, but not logits convex when $\Psi_{t,i} < 0$ (Proof. See Appendix I.2).*

**Proposition 5.** *The policy gradient loss function with importance sampling $\mathcal{L}^t_{\text{PG-IS}}$, as defined in Equation 4, is not logits convex at any time step (Proof. See Appendix I.3).*

Taken together, these propositions suggest that $\mathcal{L}^t_{\text{SFT}}$ promotes smooth and stable gradient behavior, whereas $\mathcal{L}^t_{\text{PG}}$ and $\mathcal{L}^t_{\text{PG-IS}}$ exhibit potential gradient instability, consistent with the oscillations observed in practice. Furthermore, by leveraging the general consequences of logits convexity, Proposition 1 addresses the issue of gradient divergence in $\mathcal{L}^t_{\text{PG}}$ during convergence, while Proposition 2 mitigates the risk of gradient magnitude spikes in $\mathcal{L}^t_{\text{PG-IS}}$. These insights motivate the design of a new RL objective that explicitly enforces logits convexity to achieve more stable training.

## 4.2 LOGITS CONVEX OPTIMIZATION

Motivated by the above analysis, we introduce *Logits Convex Optimization* (*LCO*), a training objective that stabilizes gradient dynamics in reinforcement learning. The key idea is to construct a target distribution that guides the policy model to encourage beneficial actions while suppressing undesirable ones, aligning with the core goal of policy gradient methods. Concretely, LCO minimizes the KL divergence between the policy distribution $\pi_\theta(\cdot|s_t)$ and a target distribution $\pi'(\cdot|s_t)$.

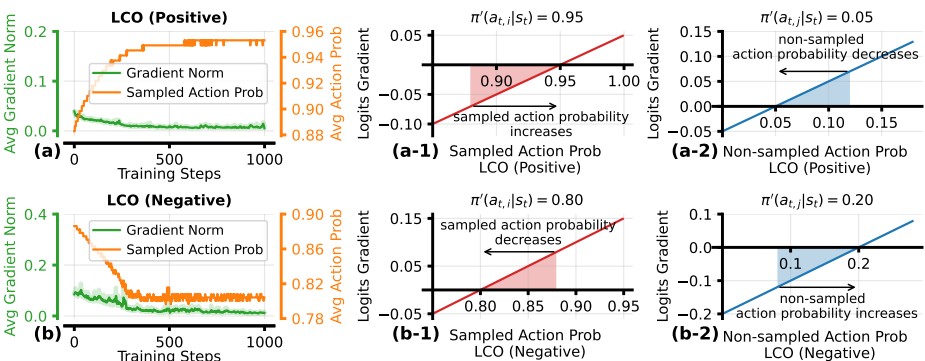

Figure 5: Training dynamics of LCO. **(a)** Gradient norm $\|\nabla_\theta \mathcal{L}_{\text{LCO}}^t\|$ for positive actions ($\Psi_{t,i} > 0$). **(b)** Gradient norm $\|\nabla_\theta \mathcal{L}_{\text{LCO}}^t\|$ for negative actions ($\Psi_{t,i} < 0$). **(a-1)** & **(b-1)** Sampled action logit gradient: $\partial \mathcal{L}_{\text{LCO}}^t / \partial z_\theta(a_{t,i}|s_t) = \pi_\theta(a_{t,i}|s_t) - \pi'(a_{t,i}|s_t)$. **(a-2)** & **(b-2)** Non-sampled action logit gradient: $\partial \mathcal{L}_{\text{LCO}}^t / \partial z_\theta(a_{t,j}|s_t) = \pi_\theta(a_{t,j}|s_t) - \pi'(a_{t,j}|s_t)$. The $\pi'(\cdot|s_t)$ is the target distribution.

**Estimation for Target Distribution**   To specify the desired update for the probability of a sampled action $a_{t,i}$, we first define the ratio $\rho_{t,i}$ of the target probability to the current policy probability:

$$\rho_{t,i} \triangleq \frac{\pi'(a_{t,i}|s_t)}{\pi_\theta(a_{t,i}|s_t)}. \tag{13}$$

If $\Psi_{t,i} > 0$, then the probability of $a_{t,i}$ should be increased, which implies $\rho_{t,i} > 1$. If $\Psi_{t,i} < 0$, then the probability of $a_{t,i}$ should be suppressed, which implies $\rho_{t,i} < 1$.

Next, we introduce a logit adjustment $\Delta z_{t,i}$ so the updated probability of $a_{t,i}$ equals $\pi'(a_{t,i}|s_t)$. For simplicity, we apply the adjustment only to the sampled action, with its target probability given by:

$$\pi'(a_{t,i}|s_t) = \frac{\exp(z_\theta(a_{t,i}|s_t) + \Delta z_{t,i})}{\sum_{k \neq i} \exp z_\theta(a_{t,k}|s_t) + \exp(z_\theta(a_{t,i}|s_t) + \Delta z_{t,i})}. \tag{14}$$

We derive $\Delta z_{t,i}$ by combining $\rho_{t,i}$ with Equation 14 (derivation in Appendix J):

$$\Delta z_{t,i} = \log \rho_{t,i} + \log \frac{1 - \pi_\theta(a_{t,i}|s_t)}{1 - \rho_{t,i}\pi_\theta(a_{t,i}|s_t)}. \tag{15}$$

For non-sampled action $a_{t,j}$, the probability is proportionally reallocated using softmax:

$$\pi'(a_{t,j}|s_t) = \frac{\exp z_\theta(a_{t,j}|s_t)}{\sum_{k \neq i} \exp z_\theta(a_{t,k}|s_t) + \exp(z_\theta(a_{t,i}|s_t) + \Delta z_{t,i})}. \tag{16}$$

By constructing the target distribution via direct logit adjustments, LCO ensures the policy updates align with the core goal of policy gradient methods while staying close to the current policy. This proximity prevents large distribution shifts and excessive updates.

**LCO Objective**   With the target distribution $\pi'(\cdot|s_t)$ defined, the LCO objective minimizes the KL divergence between $\pi_\theta(\cdot|s_t)$ and $\pi'(\cdot|s_t)$, with $|\Psi_{t,i}|$ regulating the update strength:

$$\mathcal{L}_{\text{LCO}}^t = |\Psi_{t,i}| \sum_k^{|\mathcal{A}|} \pi'(a_{t,k}|s_t) \log \frac{\pi'(a_{t,k}|s_t)}{\pi_\theta(a_{t,k}|s_t)}. \tag{17}$$

Proposition 6 establishes that minimizing $\mathcal{L}_{\text{LCO}}^t$ produces a logits-convex objective, ensuring stable gradient behavior during RL training. This objective is applicable across different RL methods.

**Proposition 6.** *The logits convex optimization loss function $\mathcal{L}_{LCO}^t$, as defined in Equation 17, is logits convex at each time step (Proof. See Appendix I.4).*

### 4.3   GRADIENT DYNAMICS OF LCO

In this section, we analyze the gradient dynamics of $\mathcal{L}_{\text{LCO}}^t$. For a logit $z_\theta(a_{t,k}|s_t)$, the gradient of $\mathcal{L}_{\text{LCO}}^t$ with respect to $z_\theta(a_{t,k}|s_t)$ is given by (see Appendix F.4 for the derivation):

$$\frac{\partial \mathcal{L}_{\text{LCO}}^t}{\partial z_\theta(a_{t,k}|s_t)} = |\Psi_{t,i}|(\pi_\theta(a_{t,k}|s_t) - \pi'(a_{t,k}|s_t)). \tag{18}$$

Since the magnitude of $\Psi_{t,i}$ does not affect our analysis, we set $|\Psi_{t,i}| = 1$ for simplicity. Figure 5 visualizes the gradient dynamics of $\mathcal{L}_{\text{LCO}}^t$. As training converges, the magnitude of the parameter gradients smoothly diminishes to zero, indicating stable gradient dynamics.

## 5 EXPERIMENTAL SETUP

**Training Data**   We first introduce the datasets utilized for RL training. We combine the original training instructions from GSM8K Cobbe et al. (2021), MATH (Hendrycks et al., 2021b), and AIME (1983–2023) to construct our RL training dataset, which contains around 20k instruction data points. We prompt DeepSeek-R1 (Guo et al., 2025) to generate responses for these instructions. From these, we randomly select 1k instructions and filter them to ensure each has a correct response, which are then used for SFT warm-up training. The remaining 19k instructions are reserved for RL training.

**Baselines**   To assess the effectiveness of our approach, we substitute the original loss functions in three widely used RL algorithms, REINFORCE/PPO/GRPO, with the proposed LCO, yielding RE-INFORCE/PPO/GRPO+LCO. Furthermore, our baselines include RFT (Yuan et al., 2023), trained solely on positive samples, and W-REINFORCE (Zhu et al., 2025b), which reduces the weighting of positive samples in REINFORCE. We also include recently prominent baselines DAPO (Yu et al., 2025), GSPO (Zheng et al., 2025), and CISPO (Chen et al., 2025a). To ensure consistency, all baseline settings adhere to the configurations recommended in their original papers.

**RL Training**   To achieve comprehensive validation across models with varying foundational capabilities, we utilize Qwen-2.5-7B, known for its strong performance, alongside the less capable Llama-2-7B in our experiments. We also incorporate the larger-scale Qwen-2.5-32B to investigate the impact of model size. Following the setting in Guo et al. (2025), we assign a rule-based reward of $+1$ for correct responses and $-1$ for incorrect ones. Before RL training, we perform warm-up SFT training on the policies to enhance their initial reasoning capabilities. For the LCO methods, we set the learning rate to 5e-6 to ensure effective training. We treat $\rho_{t,i}$ as a hyperparameter and adjust it based on the polarity of $\Psi_{t,i}$. Specifically, we set $\rho_{t,i} = 1.8$ when $\Psi_{t,i} > 0$, and $\rho_{t,i} = 0.9$ when $\Psi_{t,i} < 0$. The experimental justification for this hyperparameter selection is provided in Appendix D.1. Additional experimental configurations are provided in Appendix C.

**Evaluation Tasks**   For mathematical reasoning evaluation, we assess models on benchmarks of varying difficulty. The more capable Qwen-2.5-7B is tested on challenging tasks, including MATH500, AMC23, MinervaMath (Lewkowycz et al., 2022), OlympiadBench (He et al., 2024), OmniMath (Gao et al., 2024), and AIME2024/2025. In contrast, the less capable Llama-2-7B is evaluated on simpler tasks such as GSM8K, SVAMP (Patel et al., 2021), ASDiv (Miao et al., 2021), and MultiArith (Koncel-Kedziorski et al., 2016). To evaluate generalization beyond mathematical reasoning, we conduct experiments on out-of-distribution tasks. This includes the complex reasoning task BBH (Suzgun et al., 2022) and the multi-task language understanding benchmarks MMLU (Hendrycks et al., 2021a), MMLU-Pro (Wang et al., 2024), and MMLU-Redux (Gema et al., 2025).

## 6 RESULTS AND ANALYSIS

### 6.1 MAIN RESULTS

**Mathematical Reasoning**   We present results on math reasoning tasks in Table 1 and Table 2, using Pass@1 and Pass@8 as the evaluation metrics. Compared to the baseline methods, (e.g., RFT, W-REINFORCE, DAPO, GSPO, and CISPO), the LCO series, which leverages Qwen-2.5-7B as the backbone, achieves improved performance across most benchmarks, including MATH500, AMC23, MinervaMath, and OmniMath. Specifically, REINFORCE+LCO achieves the highest Pass@1 scores on MATH500 (64.80%) and OmniMath (17.21%), as well as the best Pass@8 scores on MinervaMath (31.62%). Similarly, GRPO+LCO demonstrates exceptional performance, achieving the highest Pass@1 on MinervaMath (23.16%) and tying for the best Pass@1 on OmniMath (17.21%). Furthermore, PPO+LCO achieves the best Pass@1 on AMC23 (47.50%), showcasing the versatility of LCO in enhancing performance across various RL settings.

Table 1: Main results of Qwen-2.5-7B on challenging mathematical reasoning tasks, aligned with its capabilities. Best performances are shown in **bold**, while suboptimal ones are underlined.

| Methods | MATH500 | | AIME2024 | | AIME2025 | | AMC23 | | MinervaMath | | OlympiadBench | | OmniMath | |
| --- | --- | --- | --- | --- | --- | --- | --- | --- | --- | --- | --- | --- | --- | --- |
| | Pass@1 | Pass@8 | Pass@1 | Pass@8 | Pass@1 | Pass@8 | Pass@1 | Pass@8 | Pass@1 | Pass@8 | Pass@1 | Pass@8 | Pass@1 | Pass@8 |
| SFT | 51.80 | 74.60 | 3.33 | 6.67 | 3.33 | 3.33 | 27.50 | 62.50 | 14.34 | 29.78 | 15.58 | 29.53 | 13.46 | 24.89 |
| RFT | 60.40 | 75.80 | 10.00 | 13.33 | 3.33 | 10.00 | 30.00 | 60.00 | 16.91 | 29.78 | 17.80 | 31.60 | 16.51 | 26.40 |
| W-REINFORCE | 56.40 | 77.00 | 3.33 | 10.00 | 3.33 | 10.00 | 25.00 | 70.00 | 14.34 | 25.74 | 15.43 | 29.23 | 13.75 | 25.52 |
| DAPO | 59.20 | 77.60 | 3.33 | 10.00 | **6.67** | 10.00 | 36.00 | 62.50 | 17.71 | 30.15 | 15.00 | 32.94 | 14.18 | 25.52 |
| GSPO | 61.60 | **79.20** | 6.67 | **16.67** | 3.33 | 6.67 | 30.00 | **72.50** | 16.54 | 29.41 | 18.55 | 34.42 | 15.94 | 26.81 |
| CISPO | 59.60 | 78.60 | 6.67 | 13.33 | **6.67** | 6.67 | 30.00 | 65.00 | 18.38 | 29.78 | 16.91 | 33.09 | 15.42 | **26.94** |
| REINFORCE | 58.60 | 76.40 | 6.67 | 13.33 | 3.33 | 3.33 | 42.50 | 62.50 | 16.91 | 28.68 | 17.80 | 33.28 | 15.09 | 25.81 |
| REINFORCE+LCO | **64.80** | 78.00 | 13.33 | 13.33 | **6.67** | 10.00 | 40.00 | 65.00 | 19.12 | **31.62** | 21.07 | 33.38 | **17.21** | 26.54 |
| PPO | 56.40 | 74.80 | 6.67 | 6.67 | 3.33 | 6.67 | 32.50 | **72.50** | 15.81 | 28.31 | 14.39 | 32.49 | 14.36 | 26.13 |
| PPO+LCO | 62.80 | 74.40 | 10.00 | 13.33 | 3.33 | 10.00 | **47.50** | 67.50 | 17.65 | 28.31 | 19.88 | 30.91 | 16.92 | 24.13 |
| GRPO | 58.80 | 74.40 | 6.67 | **16.67** | 3.33 | 10.00 | 34.40 | 67.50 | 16.13 | 30.51 | 16.17 | 30.42 | 13.96 | 25.09 |
| GRPO+LCO | 64.60 | 72.80 | 10.00 | **16.67** | **6.67** | 10.00 | 45.00 | 65.00 | **23.16** | 26.10 | **21.07** | 28.34 | **17.21** | 23.13 |

Compared to REINFORCE, PPO, and GRPO, incorporating LCO shows consistent advantages. Specifically, relative to REINFORCE, REINFORCE+LCO achieves improvements of 6.20 and 6.66 points in Pass@1 scores on MATH500 and AIME2024, respectively. Similarly, PPO+LCO and GRPO+LCO outperform their original algorithms by 15.00 and 7.03 points in Pass@1 scores on AMC23 and MinervaMath,

Table 2: Results of Llama-2-7B on simpler math reasoning tasks, aligned with its capabilities. Best performances are shown in **bold**, while suboptimal ones are underlined.

| Methods | GSM8K | | SVAMP | | ASDiv | | MultiArith | |
|---|---|---|---|---|---|---|---|---|
| | Pass@1 | Pass@8 | Pass@1 | Pass@8 | Pass@1 | Pass@8 | Pass@1 | Pass@8 |
| SFT | 20.47 | 58.53 | 33.60 | 78.00 | 33.87 | 78.22 | 48.89 | 98.33 |
| RFT | 29.11 | 59.74 | 46.50 | 82.30 | 53.28 | 79.56 | 75.56 | 98.89 |
| W-REINFORCE | 24.47 | 58.23 | 35.90 | 78.40 | 39.87 | 78.79 | 56.67 | 98.89 |
| DAPO | 24.72 | 62.32 | 40.80 | 79.90 | 45.00 | 79.22 | 61.11 | 97.22 |
| GSPO | 25.85 | 64.90 | 40.30 | 82.60 | 48.30 | 80.56 | 62.78 | 98.89 |
| CISPO | 25.25 | 62.62 | 41.20 | 81.60 | 45.96 | 80.42 | 64.44 | 99.44 |
| REINFORCE | 24.41 | 56.79 | 35.30 | 74.00 | 38.04 | 76.76 | 62.22 | 98.33 |
| REINFORCE+LCO | 32.45 | 65.88 | 48.40 | 84.30 | 55.05 | 80.85 | 80.00 | 99.44 |
| PPO | 25.92 | 56.56 | 34.00 | 75.90 | 41.14 | 79.78 | 57.78 | 99.44 |
| PPO+LCO | 34.34 | 65.43 | 47.60 | 79.60 | 56.25 | 79.25 | 81.67 | 97.78 |
| GRPO | 26.08 | 62.47 | 38.40 | 81.70 | 45.62 | 80.71 | 59.44 | 98.89 |
| GRPO+LCO | 35.25 | 61.22 | 46.60 | 81.40 | 55.58 | 74.40 | 84.44 | 99.44 |

respectively. Even with the less capable Llama-2-7B backbone, LCO-based methods outperform their counterparts. For example, GRPO+LCO achieves Pass@1 score gains of 9.17 and 25.00 points over the GRPO on GSM8K and MultiArith, respectively. Likewise, REINFORCE+LCO and PPO+LCO deliver improvements of 13.1 and 15.11 points on SVAMP and ASDiv, respectively, underscoring the effectiveness of LCO across diverse foundational models with varying capabilities.

**Multitask Understanding & Complex Reasoning** We evaluate the out-of-distribution (OOD) performance of the LCO methods on multitask language understanding benchmarks, including MMLU, MMLU-Pro, and MMLU-Redux, as well as complex reasoning task BBH. As detailed in Table 3, LCO-based methods exhibit superior accuracy compared to their counterparts. GRPO+LCO achieves the highest accuracy of

Table 3: Results for Qwen-2.5-7B on out-of-distribution tasks. Best performances are shown in **bold**, while suboptimal ones are underlined.

| Methods | MMLU | MMLU-Pro | MMLU-Redux | BBH |
|---|---|---|---|---|
| REINFORCE | 73.96 | 43.29 | 70.93 | 67.31 |
| REINFORCE+LCO | 74.26 | 49.19 | 68.48 | 68.12 |
| PPO | 72.57 | 41.42 | 67.16 | 64.98 |
| PPO+LCO | 73.46 | 49.49 | 70.35 | 66.29 |
| GRPO | 71.00 | 40.90 | 67.15 | 67.64 |
| GRPO+LCO | 75.12 | 50.37 | 69.83 | 67.69 |

75.12% and 50.37% on MMLU and MMLU-Pro, respectively, outperforming GRPO's 71.00% and 40.90%. Additionally, REINFORCE+LCO and PPO+LCO achieve 68.12% and 70.35% accuracy on BBH and MMLU-Redux, compared to 67.31% and 67.16% for REINFORCE and PPO, respectively. These results highlight the strong OOD generalization and robustness of LCO-based approaches.

## 6.2 TRAINING DYNAMICS ANALYSIS

To investigate how LCO stabilizes the RL training process, we compare the training dynamics of $\mathcal{L}_{\text{PG-IS}}^t$ and $\mathcal{L}_{\text{LCO}}^t$, which are both implemented on top of the REINFORCE algorithm as the base RL framework. Additionally, the clipping mechanism is applied to $\mathcal{L}_{\text{PG-IS}}^t$. Unless specified otherwise, the experimental settings for $\mathcal{L}_{\text{PG-IS}}^t$ and $\mathcal{L}_{\text{LCO}}^t$ are kept consistent throughout the following sections. The training dynamics of $\mathcal{L}_{\text{PG-IS}}^t$ and $\mathcal{L}_{\text{LCO}}^t$ are presented in Figure 6. Additionally, the training dynamics of $\mathcal{L}_{\text{PG}}^t$ are provided in Figure 9 in the Appendix.

**Gradient Norms** The gradient norm dynamics of $\mathcal{L}_{\text{PG-IS}}^t$ and $\mathcal{L}_{\text{LCO}}^t$ are illustrated in Figures 6(a) and (b). As training progresses, $\mathcal{L}_{\text{PG-IS}}^t$ remains relatively stable during the early stages but begins to oscillate after approximately 6K steps. In contrast, the gradient norm of $\mathcal{L}_{\text{LCO}}^t$ consistently decreases throughout the entire training process. Similar trends are observed for both positive and negative gradients, as shown in Figures 6(a-1) and (b-1). Here, the positive gradient reflects contributions from action gradients where $\Psi_{t,i} > 0$, while the negative gradient corresponds to $\Psi_{t,i} < 0$. These results indicate that LCO effectively smooths the gradients, promoting stable training.

**Entropy and Action Probabilities** We further analyze the dynamics of policy entropy and sampled action probabilities, which directly reflect the exploration capability and expected behavior of policy. As shown in Figure 6 (a-2), $\mathcal{L}_{\text{PG-IS}}^t$ exhibits a sharp drop in the sampled action probabilities and an entropy explosion during later training stages, aligning with oscillations in the gradient norms. This indicates increased uncertainty in the output of policy, confirming the occurrence of collapse phenomenon. However, $\mathcal{L}_{\text{LCO}}^t$ (b-2) maintains stable entropy and action probabilities, preserving exploration capacity while ensuring effective policy optimization.

**Evaluation Results** We evaluate the performance of $\mathcal{L}_{\text{PG-IS}}^t$ and $\mathcal{L}_{\text{LCO}}^t$ on the MATH500 test set during training. As shown in Figure 6(a-3), $\mathcal{L}_{\text{PG-IS}}^t$ experiences a performance drop in the later training stages due to the training collapse. In contrast, $\mathcal{L}_{\text{LCO}}^t$ exhibits steady performance improvements and ultimately outperforms $\mathcal{L}_{\text{PG-IS}}^t$ in terms of Pass@1 score (Figure 6(b-3)). This finding demonstrates that $\mathcal{L}_{\text{LCO}}^t$ enhances policy performance while maintaining training stability.

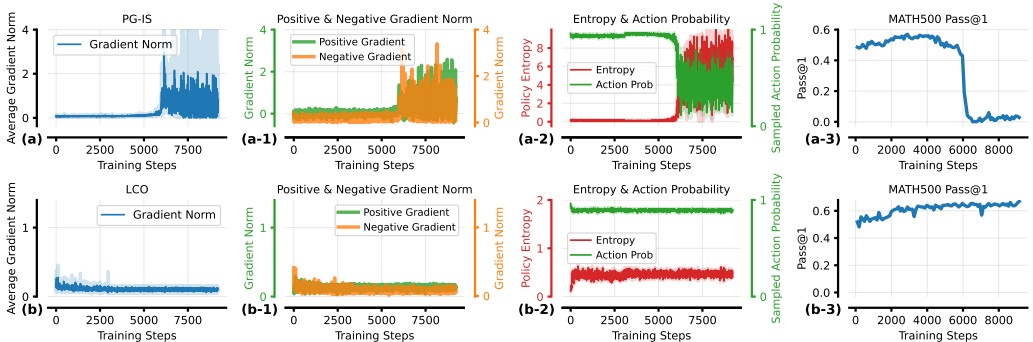

Figure 6: Training dynamics of $\mathcal{L}_{\text{PG-IS}}^t$ and $\mathcal{L}_{\text{LCO}}^t$. The analysis covers four key metrics: gradient norms, policy entropy, action probabilities, and evaluation performance.

### 6.3 TRAINING WITH DIFFERENT LEARNING RATES

We evaluate the performance trajectories of $\mathcal{L}_{\text{PG}}^t$, $\mathcal{L}_{\text{PG-IS}}^t$ and $\mathcal{L}_{\text{LCO}}^t$ under different learning rates across training iterations. All three methods are implemented on top of the REINFORCE framework. As shown in Figure 7, performance is highly sensitive to the learning rate in non-convex optimization methods such as PG and PG-IS, where higher learning rates often lead to unstable training dynamics. In contrast, LCO exhibits robust adaptability across different learning rates, achieving stable improvements and reaching its best performance with a larger learning rate.

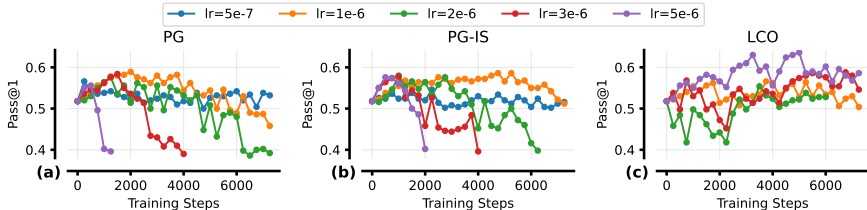

Figure 7: MATH500 Pass@1 across learning rates during training of $\mathcal{L}_{\text{PG}}^t$, $\mathcal{L}_{\text{PG-IS}}^t$ and $\mathcal{L}_{\text{LCO}}^t$. PG and PG-IS become unstable at higher rates, whereas LCO remains stable with increasing performance.

### 6.4 PG-IS PERFORMANCE ON LOW-PROBABILITY POSITIVE SAMPLES

Based on Figure 4(a-1), our analysis suggests that training on low-probability positive actions may also destabilize $\mathcal{L}_{\text{PG-IS}}^t$. To verify this, during the policy rollout phase, we selected the top 50% of positive samples with the highest perplexity. As illustrated in Figure 8, this leads to unstable training dynamics, with oscillating gradient norms and fluctuating action probabilities. Consequently, entropy oscillates and evaluation performance declines during the later stages of training. These results show that low-probability positive samples destabilize learning. Therefore, the LCO method should be applied to positive-sample gradients alongside negative samples to maintain stability.

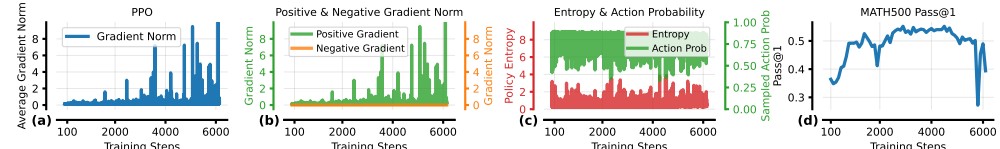

Figure 8: Training dynamics of $\mathcal{L}_{\text{PG-IS}}^t$ on positive samples with low probabilities. Training collapse still occurred in the later stages, indicating that applying LCO to positive samples is also necessary.

## 7 CONCLUSION

In this work, we analyzed the sources of reinforcement learning (RL) instability in LLMs and identified *logits convexity* as a key property underlying stable gradient behavior. We demonstrated that while supervised fine-tuning exhibits inherent stability due to logits convexity, standard RL objectives lack this property, leading to large gradient fluctuations and training collapse. Leveraging this insight, we proposed Logits Convex Optimization (LCO), a RL objective that preserves logits convexity, mitigates sudden gradient spikes, and can be seamlessly integrated into existing RL algorithms. Empirical results show that LCO delivers consistently stable training and improved performance across both reasoning and non-reasoning tasks. Our findings provide both a theoretical explanation for RL instability and a practical framework for more reliable optimization of LLMs.

## ETHICS STATEMENT

Our work focuses on improving the training stability of reinforcement learning algorithms, which we believe does not inherently raise significant ethical concerns. We have taken care to ensure that our methodologies and applications align with responsible research practices. The datasets used in this study are publicly available and widely recognized within the research community, and we have verified that their use complies with all associated terms and conditions. Additionally, we have adhered to all relevant legal and ethical standards throughout the research process. Finally, we confirm that no conflicts of interest or sponsorships have influenced the outcomes of this work.

## REPRODUCIBILITY STATEMENT

Full experimental details, including data processing and training configuration, are provided in Section 5 and Appendix C. The implemented code and data are included in the supplementary materials and will be made publicly available. Proofs for the core theoretical results (Proposition 1 and Proposition 2) are provided in Appendix G and Appendix H, respectively. These proofs assume the loss function is twice-differentiable over the real numbers.

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

## A    STATEMENT ON THE USE OF LARGE LANGUAGE MODELS

In this work, we utilize large language models solely for the purpose of polishing the manuscript. Specifically, they are employed to improve clarity and precision of phrasing, ensure grammatical correctness and spelling accuracy, and provide suggestions to enhance overall coherence and readability. The core research problem, conceptual framework, methodologies, analysis, and results are entirely developed by the authors. Our use of LLMs is strictly confined to improving the efficiency and quality of academic writing without influencing the intellectual contributions of this work.

## B    RELATED WORK

Recent research in reinforcement learning has increasingly focused on improving the stability of policy training. These efforts can be broadly categorized into three groups.

The **first category** aims to reduce the variance or bias in advantage estimation. A seminal work in this line is the GAE (Schulman et al., 2015b), which combines Monte Carlo returns and a value model to balance bias and variance. Extending GAE, VC-PPO (Yuan et al., 2025) identifies a failure mode where the value model exhibits bias during training, resulting in large errors in advantage estimation. To address this, they propose a pretraining procedure for the value model, and decouple the $\lambda$ in GAE for the policy and value model computations. Zhang et al. (2025) identify outliers caused by the imbalance in the advantage distribution. They propose StableReinforce, which applies an advantage filter to retain only those advantages that fall within three standard deviations for stable training. By simplifying the advantage estimation process, RLOO (Ahmadian et al., 2024) employs a leave-one-out baseline across multiple completions to produce an unbiased advantage estimate for a single prompt. Similarly, Shao et al. (2024) introduce GRPO, which standardizes sequence-level rewards by subtracting the mean and dividing by the standard deviation, thereby reducing bias and variance. Extending GRPO, Yu et al. (2025) propose DAPO, which re-weights token-level losses to prevent longer responses from being underrepresented in gradient updates.

The **second category** stabilizes training by constraining policy updates through a Kullback-Leibler (KL) divergence penalty relative to a reference model. For example, TRPO (Schulman et al., 2015a) aims to find a policy that increases the probability of advantageous actions while limiting the divergence from the previous policy using a KL constraint, ensuring stable training. Building upon PPO, Ouyang et al. (2022); Hu et al. (2025) add a token-level KL penalty to the reward function, which constrains the policy at each generation step to remain close to the reference SFT model. GRPO (Shao et al., 2024) modifies this approach by applying the KL constraint directly to the policy loss rather than the reward, which allows for more targeted optimization. KL-Cov (Cui et al., 2025) advances this idea by analyzing policy entropy, showing that entropy change is driven by the covariance between action probabilities and advantages, and applying KL penalties selectively to high-covariance tokens to prevent entropy collapse and improve stability.

The **third category** employs clipping mechanisms to stabilize policy updates. PPO and GRPO constrain the importance sampling ratio between current and previous policies within fixed upper and lower bounds to prevent excessively large policy updates. However, such bounds can limit training efficiency and unduly constrain specific updates. To address this, DAPO (Yu et al., 2025) proposes a decoupled clip-higher method that relaxes the upper clipping bound to improve training efficiency while maintaining stability. Building upon the same idea, DCPO (Yang et al., 2025b) addresses the limitation in DAPO, where the same clip range is set for different positions. It further introduces a dynamic clipping method that adaptively adjusts the clipping bounds based on the token-specific probabilities from previous iterations, thereby mitigating the drawbacks of fixed clipping bounds. Chen et al. (2025a) identify a key limitation in PPO/GRPO: clipping can prematurely drop high-advantage tokens from contributing to off-policy gradients. They introduce CISPO, which clips importance sampling weights without clipping token updates to stabilize training. Extending this covariance analysis, Cui et al. (2025) propose Clip-Cov, which applies clipping selectively to updates on high-covariance tokens to further enhance training stability.

Unlike previous work, our study is inspired by the stable training of SFT and provides a theoretical analysis of RL instability from a gradient perspective. We identify a property, termed *logits convexity*, which induces smoother gradient updates during optimization and ensures more stable RL training. Building on this insight, we propose a simple yet effective policy optimization strategy.

Table 4: Ablation study on KL divergence variation for $\rho_{t,i}$ in LCO methods. Configurations that achieve the closest match to the KL divergence (between the policy distributions before and after one update step) compared to their corresponding original methods are highlighted in **bold**.

| Method | $\rho_{t,i}$ $\Psi_{t,i}>0$ | $\Psi_{t,i}<0$ | KL | $\|\Delta\text{KL}\|$ | Method | $\rho_{t,i}$ $\Psi_{t,i}>0$ | $\Psi_{t,i}<0$ | KL | $\|\Delta\text{KL}\|$ | Method | $\rho_{t,i}$ $\Psi_{t,i}>0$ | $\Psi_{t,i}<0$ | KL | $\|\Delta\text{KL}\|$ |
|---|---|---|---|---|---|---|---|---|---|---|---|---|---|---|
| REINFORCE | N/A | N/A | 0.0264 | 0.0000 | PPO | N/A | N/A | 0.0216 | 0.0000 | GRPO | N/A | N/A | 0.0324 | 0.0000 |
| REINFORCE+LCO | 1.8 | 0.8 | 0.0336 | 0.0072 | PPO+LCO | 1.8 | 0.8 | 0.0314 | 0.0098 | GRPO+LCO | 1.8 | 0.8 | 0.0487 | 0.0163 |
| | 1.8 | 0.9 | 0.0273 | **0.0009** | | 1.8 | 0.9 | 0.0241 | **0.0025** | | 1.8 | 0.9 | 0.0368 | 0.0044 |
| | 1.8 | 0.95 | 0.0134 | 0.0130 | | 1.8 | 0.95 | 0.0096 | 0.0120 | | 1.8 | 0.95 | 0.0216 | 0.0108 |
| | 1.9 | 0.9 | 0.0281 | 0.0017 | | 1.9 | 0.9 | 0.0263 | 0.0047 | | 1.9 | 0.9 | 0.0396 | 0.0072 |
| | 1.7 | 0.9 | 0.0236 | 0.0028 | | 1.7 | 0.9 | 0.0183 | 0.0033 | | 1.7 | 0.9 | 0.0314 | **0.0010** |

## C  ADDITIONAL EXPERIMENTAL SETUP

To initialize the policies with basic instruction-following and reasoning capabilities while avoiding overfitting, we perform SFT warm-up training for only 1 epoch. For RL methods, we set the rollout batch size to 2,048, with 4 responses generated per instruction. The update batch size is set to 256, following Zhu et al. (2025b). A sampling temperature of 0.6 and a top-$p$ value of 0.95 are consistently applied across all policies to control the diversity and quality of generated responses.

To ensure reproducibility, all baseline configurations strictly follow the settings in their original papers. These configurations are further supplemented by the default parameters from the TRL repository[1], a widely used library for training language models with reinforcement learning.

All experiments utilize bfloat16 precision to optimize memory usage and computational efficiency. Evaluations are performed in a zero-shot setting. Consistent with training, a sampling temperature of 0.6 and a top-$p$ value of 0.95 are used during evaluation, as recommended by Guo et al. (2025).

## D  ADDITIONAL EXPERIMENTAL RESULTS

### D.1  ABLATION STUDY ON $\rho_{t,i}$

In this section, we present a comprehensive ablation study on the hyperparameter $\rho_{t,i}$ in the LCO framework, using Qwen-2.5-7B as the base model. Individually tuning $\rho_{t,i}$ at each time step $t$ is computationally prohibitive, so we instead treat $\rho_{t,i}$ across all steps as a unified parameter. For $\Psi_{t,i}>0$, we consider values greater than 1, and for $\Psi_{t,i}<0$, values smaller than 1. Accordingly, we search within constrained ranges: $\rho_{t,i} \in \{1.2, 1.7, 1.8, 1.9\}$ when $\Psi_{t,i}>0$ and $\rho_{t,i} \in \{0.7, 0.8, 0.9, 0.95\}$ when $\Psi_{t,i}<0$, to identify the optimal update magnitude. As

Table 5: Ablation study of $\rho_{t,i}$ in the REINFORCE+LCO method.

| Method | $\rho_{t,i}$ $\Psi_{t,i}>0$ | $\Psi_{t,i}<0$ | Pass@1 |
|---|---|---|---|
| REINFORCE+LCO | 1.2 | 0.7 | 55.20 |
| | 1.2 | 0.8 | 58.60 |
| | 1.7 | 0.8 | 61.20 |
| | 1.8 | 0.8 | 63.60 |
| | 1.8 | 0.9 | **64.80** |
| | 1.8 | 0.95 | 64.00 |
| | 1.9 | 0.9 | 63.40 |
| | 1.7 | 0.9 | 62.40 |

shown in Table 5, the configuration $\rho_{t,i}=1.8$ for $\Psi_{t,i}>0$ and $\rho_{t,i}=0.9$ for $\Psi_{t,i}<0$ consistently delivers the best performance, achieving a Pass@1 of 64.80% on the evaluation set. This setting strikes a balance: $\rho_{t,i}=1.8$ amplifies beneficial actions, while $\rho_{t,i}=0.9$ suppresses undesirable ones without introducing gradient instability.

We further evaluate the KL divergence between policy distributions before and after a single training update for REINFORCE, PPO, GRPO, and their LCO-augmented counterparts. The objective is to find $\rho_{t,i}$ values that align the KL divergence of LCO-augmented methods with their baselines. As reported in Table 4, the same configuration ($\rho_{t,i}=1.8$ for $\Psi_{t,i}>0$, $\rho_{t,i}=0.9$ for $\Psi_{t,i}<0$) in REINFORCE+LCO and PPO+LCO yields the closest KL divergence to the original methods. Considering both Pass@1 and KL divergence, we adopt this configuration as the default for LCO.

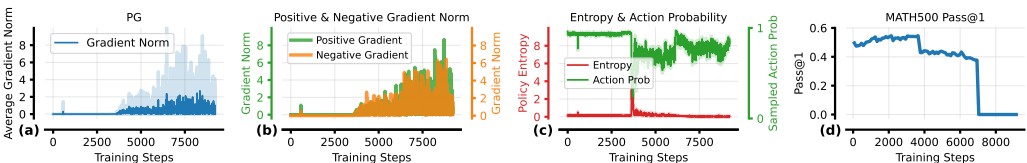

Figure 9: Training dynamics of $\mathcal{L}_{\text{PG}}^t$. The analysis covers four key metrics: gradient norms, policy entropy, action probabilities, and evaluation performance.

---

[1] https://github.com/huggingface/trl

## D.2 Training Dynamics Analysis for PG

As an extensive experiment of Section 6.2, we show the training dynamic of $\mathcal{L}_{\text{PG}}^t$ in Figure 9. Following the settings of $\mathcal{L}_{\text{PG-IS}}^t$ and $\mathcal{L}_{\text{LCO}}^t$, $\mathcal{L}_{\text{PG}}^t$ is also implemented on top of the REINFORCE algorithm as the base RL framework.

The gradient norm dynamics of $\mathcal{L}_{\text{PG}}^t$ are shown in Figure 9(a). As training progresses, the gradient norm diverges significantly, with similar trends observed for both positive and negative gradients, as illustrated in Figure 9(b). Additionally, we analyze the dynamics of policy entropy and sampled action probabilities. As depicted in Figure 9(c), $\mathcal{L}_{\text{PG}}^t$ exhibits a sharp decline in sampled action probabilities at approximately 4K steps, coinciding with oscillations in the gradient norms. Concurrently, policy entropy fluctuates in a similar manner. Furthermore, we evaluate $\mathcal{L}_{\text{PG}}^t$'s performance on the MATH500 test set during training. As shown in Figure 9(d), a performance drop is observed around 4K steps, which aligns with the oscillations in the gradient norms.

Table 6: Additional results for Qwen-2.5-7B on challenging mathematical reasoning tasks, aligned with its capabilities. Best performances are shown in **bold**, while suboptimal ones are underlined.

| Methods | MATH500 | | AIME2024 | | AIME2025 | | AMC23 | | MinervaMath | | OlympiadBench | | OmniMath | |
|---|---|---|---|---|---|---|---|---|---|---|---|---|---|---|
| | Pass@1 | Pass@8 | Pass@1 | Pass@8 | Pass@1 | Pass@8 | Pass@1 | Pass@8 | Pass@1 | Pass@8 | Pass@1 | Pass@8 | Pass@1 | Pass@8 |
| NFT | 54.80 | 75.80 | 3.33 | 10.00 | 3.33 | 10.00 | 27.50 | 65.00 | 12.87 | 30.15 | 16.02 | 30.86 | 13.82 | 25.93 |
| NSR | 53.20 | 75.80 | 0.00 | 6.67 | 3.33 | **13.33** | 22.50 | 57.50 | 15.07 | 26.84 | 16.77 | 30.56 | 14.00 | 25.56 |
| OREAL | 56.80 | **78.40** | 6.67 | 13.33 | 0.00 | 10.00 | 32.50 | 62.50 | 15.81 | 27.94 | 15.13 | 31.01 | 14.32 | 26.20 |
| RLOO | 57.60 | 78.00 | 3.33 | **20.00** | 3.33 | 6.67 | 32.40 | **67.50** | 15.24 | 29.78 | 17.95 | 31.75 | 15.15 | **26.54** |
| REINFORCE+LCO | **64.80** | 78.00 | **13.33** | 13.33 | **6.67** | 10.00 | 40.00 | 65.00 | 19.12 | **31.62** | 21.07 | **33.38** | **17.21** | **26.54** |
| PPO+LCO | 62.80 | 74.40 | 10.00 | 13.33 | 3.33 | 10.00 | **47.50** | **67.50** | 17.65 | 28.31 | 19.88 | 30.91 | 16.92 | 24.13 |
| GRPO+LCO | 64.60 | 72.80 | 10.00 | 16.67 | **6.67** | 10.00 | 45.00 | 65.00 | **23.16** | 26.10 | **21.07** | 28.34 | **17.21** | 23.13 |

Table 7: Additional results for Llama-2-7B on simpler mathematical reasoning tasks, aligned with its capabilities. Best performances are shown in **bold**, while suboptimal ones are underlined.

| Methods | GSM8K | | SVAMP | | ASDiv | | MultiArith | |
|---|---|---|---|---|---|---|---|---|
| | Pass@1 | Pass@8 | Pass@1 | Pass@8 | Pass@1 | Pass@8 | Pass@1 | Pass@8 |
| NFT | 26.62 | 56.86 | 34.70 | 78.90 | 42.03 | 79.32 | 57.78 | 97.22 |
| NSR | 21.38 | 57.24 | 32.90 | 76.80 | 37.48 | 79.61 | 58.89 | 97.22 |
| OREAL | 26.99 | 61.03 | 38.30 | 79.40 | 42.60 | 78.55 | 58.33 | 98.33 |
| RLOO | 26.38 | 62.02 | 41.50 | 81.40 | 44.81 | **81.62** | 66.11 | 96.67 |
| REINFORCE+LCO | 32.45 | **65.88** | **48.40** | **84.30** | 55.05 | 80.85 | 80.00 | **99.44** |
| PPO+LCO | 34.34 | 65.43 | 47.60 | 79.60 | **56.25** | 79.25 | 81.67 | 97.78 |
| GRPO+LCO | **35.25** | 61.22 | 46.60 | 81.40 | 55.58 | 74.40 | **84.44** | **99.44** |

## D.3 Additional Baselines Comparison

Additionally, we compare our method against a broader set of baselines, including RLOO (Ahmadian et al., 2024), NSR (Zhu et al., 2025b), NFT (Chen et al., 2025b), and OREAL (Lyu et al., 2025). Results on the test sets of various mathematical reasoning tasks are presented in Tables 6 and 7, with Pass@1 and Pass@8 as the evaluation metrics. Whether using the more powerful Qwen-2.5-7B model or the less advanced Llama-2-7B backbone, the LCO series methods consistently enhance performance across different baselines in most mathematical reasoning tasks.

## D.4 Impact of Model Size

To investigate the impact of model size on LCO, we adopt Qwen-2.5-32B as the policy backbone and compare three training approaches: SFT, REINFORCE, and REINFORCE+LCO. Performance is evaluated using the Pass@1 metric on the MATH500 benchmark. As shown in Figure 10, LCO consistently outperforms both SFT and REINFORCE when scaling the policy model to 32B parameters. These results demonstrate the robustness and scalability of LCO, confirming its effectiveness not only for smaller models such as 7B but also for substantially larger ones.

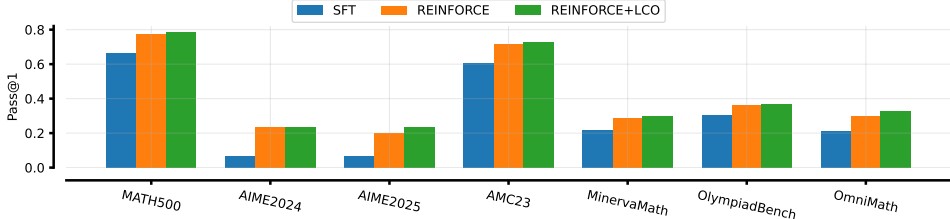

Figure 10: Performance of SFT, REINFORCE, and REINFORCE+LCO using Qwen-2.5-32B.

# E  GRADIENT OF THE SOFTMAX FUNCTION

Consider the output of the softmax function, denoted as $\pi_\theta(a_{t,i}|s_t)$, which is defined as:

$$\pi_\theta(a_{t,i}|s_t) = \frac{\exp z_\theta(a_{t,i}|s_t)}{\sum_{k'} \exp z_\theta(a_{t,k'}|s_t)}. \tag{19}$$

The gradient of the softmax function with respect to the logits $z_\theta(a_{t,k}|s_t)$ can be derived in two cases. The first case is when $i \neq k$:

$$\begin{aligned}
\frac{\partial \pi_\theta(a_{t,i}|s_t)}{\partial z_\theta(a_{t,k}|s_t)} &= -\frac{\exp z_\theta(a_{t,i}|s_t) \exp z_\theta(a_{t,k}|s_t)}{\left(\sum_{k'} \exp z_\theta(a_{t,k'}|s_t)\right)^2} \\
&= -\pi_\theta(a_{t,i}|s_t)\pi_\theta(a_{t,k}|s_t).
\end{aligned} \tag{20}$$

The second case is when $i = k$:

$$\begin{aligned}
\frac{\partial \pi_\theta(a_{t,i}|s_t)}{\partial z_\theta(a_{t,k}|s_t)} &= \frac{\exp z_\theta(a_{t,i}|s_t)}{\sum_{k'} \exp z_\theta(a_{t,k'}|s_t)} - \frac{\exp z_\theta(a_{t,i}|s_t) \exp z_\theta(a_{t,k}|s_t)}{\left(\sum_{k'} \exp z_\theta(a_{t,k'}|s_t)\right)^2} \\
&= \pi_\theta(a_{t,i}|s_t)(1 - \pi_\theta(a_{t,k}|s_t)).
\end{aligned} \tag{21}$$

To unify these two cases, we introduce the Kronecker delta function, defined as $\delta_{ik} = 1$ if $i = k$, and $\delta_{ik} = 0$ otherwise. Using this definition, the gradient can be written as:

$$\frac{\partial \pi_\theta(a_{t,i}|s_t)}{\partial z_\theta(a_{t,k}|s_t)} = \pi_\theta(a_{t,i}|s_t)(\delta_{ik} - \pi_\theta(a_{t,k}|s_t)). \tag{22}$$

# F  DERIVATION OF LOGITS GRADIENT

## F.1  LOGITS GRADIENT OF SUPERVISED FINE-TUNING

We provide the derivation for Equation 6. The SFT loss at time step $t$ is:

$$\mathcal{L}_{\text{SFT}}^t = -\log \pi_\theta(a_{t,i}|s_t). \tag{23}$$

For a logit $z_\theta(a_{t,k}|s_t)$, we compute the partial derivative using Equation 22:

$$\begin{aligned}
\frac{\partial \mathcal{L}_{\text{SFT}}^t}{\partial z_\theta(a_{t,k}|s_t)} &= -\frac{\partial \log \pi_\theta(a_{t,i}|s_t)}{\partial \pi_\theta(a_{t,i}|s_t)} \frac{\partial \pi_\theta(a_{t,i}|s_t)}{\partial z_\theta(a_{t,k}|s_t)} \\
&= -\frac{1}{\pi_\theta(a_{t,i}|s_t)}\pi_\theta(a_{t,i}|s_t)(\delta_{ik} - \pi_\theta(a_{t,k}|s_t)) \\
&= \pi_\theta(a_{t,k}|s_t) - \delta_{ik}.
\end{aligned} \tag{24}$$

## F.2  LOGITS GRADIENT OF POLICY GRADIENT

We provide the derivation for Equation 7. The policy gradient (PG) loss at time step $t$ is:

$$\mathcal{L}_{\text{PG}}^t = -\Psi_{t,i} \log \pi_\theta(a_{t,i}|s_t). \tag{25}$$

For a logit $z_\theta(a_{t,k}|s_t)$, we compute the partial derivative using Equation 22:

$$\begin{aligned}
\frac{\partial \mathcal{L}_{\text{PG}}^t}{\partial z_\theta(a_{t,k}|s_t)} &= -\Psi_{t,i}\frac{\partial \log \pi_\theta(a_{t,i}|s_t)}{\partial \pi_\theta(a_{t,i}|s_t)} \frac{\partial \pi_\theta(a_{t,i}|s_t)}{\partial z_\theta(a_{t,k}|s_t)} \\
&= -\Psi_{t,i}\frac{1}{\pi_\theta(a_{t,i}|s_t)}\pi_\theta(a_{t,i}|s_t)(\delta_{ik} - \pi_\theta(a_{t,k}|s_t)) \\
&= \Psi_{t,i}(\pi_\theta(a_{t,k}|s_t) - \delta_{ik}).
\end{aligned} \tag{26}$$

### F.3 LOGITS GRADIENT OF POLICY GRADIENT WITH IMPORTANCE SAMPLING

We provide the derivation for Equation 8. The PG-IS loss at time step $t$ is:

$$\mathcal{L}_{\text{PG-IS}}^t = -\Psi_{t,i} \frac{\pi_\theta(a_{t,i}|s_t)}{\pi_{\theta_{\text{old}}}(a_{t,i}|s_t)}. \tag{27}$$

For a logit $z_\theta(a_{t,k}|s_t)$, we the compute partial derivative using Equation 22:

$$
\begin{aligned}
\frac{\partial \mathcal{L}_{\text{PG-IS}}^t}{\partial z_\theta(a_{t,k}|s_t)} &= -\frac{\Psi_{t,i}}{\pi_{\theta_{\text{old}}}(a_{t,i}|s_t)} \frac{\partial \pi_\theta(a_{t,i}|s_t)}{\partial z_\theta(a_{t,k}|s_t)} \\
&= \frac{\Psi_{t,i}}{\pi_{\theta_{\text{old}}}(a_{t,i}|s_t)} \pi_\theta(a_{t,i}|s_t)(\pi_\theta(a_{t,k}|s_t) - \delta_{ik}).
\end{aligned}
\tag{28}
$$

### F.4 LOGITS GRADIENT OF LCO

We provide the derivation for Equation 18. The LCO loss at time step $t$ is:

$$\mathcal{L}_{\text{LCO}}^t = |\Psi_{t,i}| \sum_{k'} \pi'(a_{t,k'}|s_t) \log \frac{\pi'(a_{t,k'}|s_t)}{\pi_\theta(a_{t,k'}|s_t)}. \tag{29}$$

For a logit $z_\theta(a_{t,k}|s_t)$, we compute the partial derivative using Equation 22:

$$
\begin{aligned}
\frac{\partial \mathcal{L}_{\text{LCO}}^t}{\partial z_\theta(a_{t,k}|s_t)} &= |\Psi_{t,i}| \sum_{k'} \left[ -\frac{\pi'(a_{t,k'}|s_t)}{\pi_\theta(a_{t,k'})} \frac{\partial \pi_\theta(a_{t,k'}|s_t)}{\partial z_\theta(a_{t,k}|s_t)} \right] \\
&= |\Psi_{t,i}| \sum_{k'} \left[ -\frac{\pi'(a_{t,k'}|s_t)}{\pi_\theta(a_{t,k'}|s_t)} \pi_\theta(a_{t,k'}|s_t)(\delta_{k'k} - \pi_\theta(a_{t,k}|s_t)) \right] \\
&= |\Psi_{t,i}| \sum_{k'} \left[ -\pi'(a_{t,k'}|s_t)(\delta_{k'k} - \pi_\theta(a_{t,k}|s_t)) \right] \\
&= |\Psi_{t,i}| \left[ -\sum_{k'} \pi'(a_{t,k'}|s_t)\delta_{k'k} + \sum_{k'} \pi'(a_{t,k'}|s_t)\pi_\theta(a_{t,k}|s_t) \right] \\
&= |\Psi_{t,i}|(\pi_\theta(a_{t,k}|s_t) - \pi'(a_{t,k}|s_t)).
\end{aligned}
\tag{30}
$$

## G PROOF OF PROPOSITION 1

In this section, we provide the proof for Proposition 1. Let $\mathcal{L} : \mathbb{R}^n \to \mathbb{R}$ be a twice-differentiable loss function that takes logits $\boldsymbol{z}_\theta \in \mathbb{R}^n$ parameterized by $\theta$ as input. According to the chain rule, the gradient norm of $\mathcal{L}$ with respect to $\theta$ is given by:

$$\|\nabla_\theta \mathcal{L}\| = \left\| \sum_i \frac{\partial \mathcal{L}}{\partial z_{\theta,i}} \nabla_\theta z_{\theta,i} \right\|. \tag{31}$$

where $z_{\theta,i}$ is the $i$-th element of $\boldsymbol{z}_\theta$. According to the triangle inequality, we have:

$$\left\| \sum_i \frac{\partial \mathcal{L}}{\partial z_{\theta,i}} \nabla_\theta z_{\theta,i} \right\| \leq \sum_i \left\| \frac{\partial \mathcal{L}}{\partial z_{\theta,i}} \nabla_\theta z_{\theta,i} \right\| = \sum_i \left| \frac{\partial \mathcal{L}}{\partial z_{\theta,i}} \right| \left\| \nabla_\theta z_{\theta,i} \right\|. \tag{32}$$

If $\mathcal{L}$ is logits convex, then by the Definition 1, $\mathcal{L}$ is convex with respect to $\boldsymbol{z}_\theta$. Let $\boldsymbol{z}_\theta^* \in \mathbb{R}^n$ be the optimal logits, and $z_{\theta,i}^*$ be the $i$-th element of $\boldsymbol{z}_\theta^*$. For each $i$, we have:

$$\lim_{z_{\theta,i} \to z_{\theta,i}^*} \left| \frac{\partial \mathcal{L}}{\partial z_{\theta,i}} \right| = 0 \Rightarrow \lim_{\boldsymbol{z}_\theta \to \boldsymbol{z}_\theta^*} \sum_i \left| \frac{\partial \mathcal{L}}{\partial z_{\theta,i}} \right| \left\| \nabla_\theta z_{\theta,i} \right\| = 0. \tag{33}$$

So by the squeeze theorem, we have the following result:

$$0 \leq \|\nabla_\theta \mathcal{L}\| \leq \sum_i \left| \frac{\partial \mathcal{L}}{\partial z_{\theta,i}} \right| \left\| \nabla_\theta z_{\theta,i} \right\| \Rightarrow \lim_{\boldsymbol{z}_\theta \to \boldsymbol{z}_\theta^*} \|\nabla_\theta \mathcal{L}\| = 0. \tag{34}$$

This completes the proof.

# H    PROOF OF PROPOSITION 2

In this section, we provide the proof for Proposition 2. Let $\mathcal{L} : \mathbb{R}^n \to \mathbb{R}$ be a twice-differentiable loss function that takes logits $\boldsymbol{z}_\theta \in \mathbb{R}^n$ parameterized by $\theta$ as input. If $\mathcal{L}$ is logits convex, by the Definition 1, $\mathcal{L}$ is convex with respect to $\boldsymbol{z}_\theta$. By the first-order characterization of convexity (Boyd & Vandenberghe, 2004), for any two vectors $\boldsymbol{z}'_\theta, \boldsymbol{z}''_\theta \in \mathbb{R}^n$, the following inequalities hold:

$$
\begin{aligned}
\mathcal{L}(\boldsymbol{z}'_\theta) &\geq \mathcal{L}(\boldsymbol{z}''_\theta) + \sum_k \frac{\partial \mathcal{L}}{\partial z''_{\theta,k}}(z'_{\theta,k} - z''_{\theta,k}), \\
\mathcal{L}(\boldsymbol{z}''_\theta) &\geq \mathcal{L}(\boldsymbol{z}'_\theta) + \sum_k \frac{\partial \mathcal{L}}{\partial z'_{\theta,k}}(z''_{\theta,k} - z'_{\theta,k}),
\end{aligned}
\tag{35}
$$

where $z_{\theta,k}$ is the $k$-th element of $\boldsymbol{z}_\theta$. We then reformulate Equation 35 as:

$$
\sum_k \frac{\partial \mathcal{L}}{\partial z'_{\theta,k}}(z'_{\theta,k} - z''_{\theta,k}) \geq \mathcal{L}(\boldsymbol{z}'_\theta) - \mathcal{L}(\boldsymbol{z}''_\theta) \geq \sum_k \frac{\partial \mathcal{L}}{\partial z''_{\theta,k}}(z'_{\theta,k} - z''_{\theta,k}).
\tag{36}
$$

To analyze the $i$-th component of the logits vector, we fix all other components of $\boldsymbol{z}'_\theta$ and $\boldsymbol{z}''_\theta$ by setting $z'_{\theta,k} = z''_{\theta,k}$, for $k \neq i$. Under this setup, Equation 36 then can be simplified to:

$$
\frac{\partial \mathcal{L}}{\partial z'_{\theta,i}}(z'_{\theta,i} - z''_{\theta,i}) \geq \frac{\partial \mathcal{L}}{\partial z''_{\theta,i}}(z'_{\theta,i} - z''_{\theta,i}).
\tag{37}
$$

For the optimal value $z^*_{\theta,i}$, where $\frac{\partial \mathcal{L}}{\partial z^*_{\theta,i}} = 0$ (for a convex function), we obtain these conditions:

$$
\begin{aligned}
\frac{\partial \mathcal{L}}{\partial z'_{\theta,i}}(z'_{\theta,i} - z^*_{\theta,i}) &\geq 0, \\
\frac{\partial \mathcal{L}}{\partial z''_{\theta,i}}(z''_{\theta,i} - z^*_{\theta,i}) &\geq 0.
\end{aligned}
\tag{38}
$$

Consider the case where $z'_{\theta,i} < z''_{\theta,i} < z^*_{\theta,i}$, using Equations 37 and 38, we have:

$$
0 \geq \frac{\partial \mathcal{L}}{\partial z''_{\theta,i}} \geq \frac{\partial \mathcal{L}}{\partial z'_{\theta,i}} \Rightarrow \left| \frac{\partial \mathcal{L}}{\partial z''_{\theta,i}} \right| \leq \left| \frac{\partial \mathcal{L}}{\partial z'_{\theta,i}} \right|.
\tag{39}
$$

Similarly, consider the case where $z'_{\theta,i} > z''_{\theta,i} > z^*_{\theta,i}$. Using Equation 37 and 38, we have:

$$
0 \leq \frac{\partial \mathcal{L}}{\partial z''_{\theta,i}} \leq \frac{\partial \mathcal{L}}{\partial z'_{\theta,i}} \Rightarrow \left| \frac{\partial \mathcal{L}}{\partial z''_{\theta,i}} \right| \leq \left| \frac{\partial \mathcal{L}}{\partial z'_{\theta,i}} \right|.
\tag{40}
$$

Combining both of the above cases, when $z'_{\theta,i}$ and $z''_{\theta,i}$ lie on the same side of the optimal value $z^*_{\theta,i}$ and $|z''_{\theta,i} - z^*_{\theta,i}| < |z'_{\theta,i} - z^*_{\theta,i}|$, the gradient magnitudes satisfy the following relationship:

$$
\left| \frac{\partial \mathcal{L}}{\partial z''_{\theta,i}} \right| \leq \left| \frac{\partial \mathcal{L}}{\partial z'_{\theta,i}} \right|.
\tag{41}
$$

This completes the proof.

# I    PROOF OF LOGITS CONVEXITY

## I.1    LOGITS CONVEXITY OF SFT LOSS

According to Equation 24, the partial derivative of $\mathcal{L}^t_{\text{SFT}}$ with respect to logit $z_\theta(a_{t,k}|s_t)$ is:

$$
\frac{\partial \mathcal{L}^t_{\text{SFT}}}{\partial z_{\theta,k}} = \pi_{\theta,k} - \delta_{ik},
\tag{42}
$$

with $z_\theta(a_{t,k}|s_t)$ simplified as $z_{\theta,k}$, and $\pi_\theta(a_{t,k}|s_t)$ simplified as $\pi_{\theta,k}$. The second derivative is:

$$\frac{\partial^2 \mathcal{L}_{\text{SFT}}^t}{\partial z_{\theta,k}\partial z_{\theta,k'}} = \frac{\partial(\pi_{\theta,k} - \delta_{ik})}{\partial z_{\theta,k'}} = \pi_{\theta,k}(\delta_{kk'} - \pi_{\theta,k'}). \tag{43}$$

To prove the convexity of the logits, we derive the Hessian matrix $\boldsymbol{H}$ of $\mathcal{L}_{\text{SFT}}^t$ with respect to the logits and check if $\boldsymbol{H}$ is positive semi-definite. The Hessian is a square matrix composed of second-order partial derivatives of the loss $\mathcal{L}_{\text{SFT}}^t$ with respect to the logits:

$$\boldsymbol{H}_{k,k'} = \frac{\partial^2 \mathcal{L}_{\text{SFT}}^t}{\partial z_{\theta,k}\partial z_{\theta,k'}}. \tag{44}$$

According to Equation 43, the individual elements of the Hessian matrix $\boldsymbol{H}$ can be decomposed using elements of two matrices $\boldsymbol{A}$ and $\boldsymbol{B}$:

$$\boldsymbol{H}_{k,k'} = \underbrace{\pi_{\theta,k}\delta_{kk'}}_{\boldsymbol{A}_{k,k'}} - \underbrace{\pi_{\theta,k}\pi_{\theta,k'}}_{\boldsymbol{B}_{k,k'}}. \tag{45}$$

Each of the two matrices $\boldsymbol{A}$ and $\boldsymbol{B}$ can be written in a compact form. Let

$$\boldsymbol{\pi}_\theta = [\pi_{\theta,1}, \pi_{\theta,2}, \ldots, \pi_{\theta,n}]^\top \tag{46}$$

represent the probability distribution over the vocabulary at time step $t$, where $n$ is the vocabulary size. The $\boldsymbol{A}$ and $\boldsymbol{B}$ are both $n \times n$ matrices with the following structure:

$$\begin{aligned} \boldsymbol{A} &= \text{diag}(\boldsymbol{\pi}_\theta), \\ \boldsymbol{B} &= \boldsymbol{\pi}_\theta\boldsymbol{\pi}_\theta^\top, \end{aligned} \tag{47}$$

where $\text{diag}(\boldsymbol{\pi}_\theta)$ is a diagonal matrix with $\pi_{\theta,i}$ as its $i$-th diagonal entry. Then the Hessian matrix $\boldsymbol{H}$ has the following structure:

$$\boldsymbol{H} = \text{diag}(\boldsymbol{\pi}_\theta) - \boldsymbol{\pi}_\theta\boldsymbol{\pi}_\theta^\top. \tag{48}$$

To prove convexity, we need to show that the Hessian matrix $\boldsymbol{H}$ is positive semi-definite. For any random vector $\mathbf{v} \in \mathbb{R}^n$, consider the quadratic form:

$$\mathbf{v}^\top \boldsymbol{H}\mathbf{v} = \mathbf{v}^\top (\text{diag}(\boldsymbol{\pi}_\theta) - \boldsymbol{\pi}_\theta\boldsymbol{\pi}_\theta^\top)\mathbf{v}. \tag{49}$$

Expanding this:

$$\mathbf{v}^\top \boldsymbol{H}\mathbf{v} = \mathbf{v}^\top \text{diag}(\boldsymbol{\pi}_\theta)\mathbf{v} - \mathbf{v}^\top \boldsymbol{\pi}_\theta\boldsymbol{\pi}_\theta^\top \mathbf{v} = \sum_k^n \pi_{\theta,k}v_k^2 - \left(\sum_k^n \pi_{\theta,k}v_k\right)^2. \tag{50}$$

We use the Cauchy-Schwarz inequality:

$$\left(\sum_k^n u_k^2\right)\left(\sum_k^n w_k^2\right) - \left(\sum_k^n u_k w_k\right)^2 \geq 0. \tag{51}$$

Let $u_k = \sqrt{\pi_{\theta,k}}$, $w_k = v_k\sqrt{\pi_{\theta,k}}$, and substitute into Equation 51 and have:

$$\mathbf{v}^\top \boldsymbol{H}\mathbf{v} = \sum_k^n \pi_{\theta,k}v_k^2 - \left(\sum_k^n \pi_{\theta,k}v_k\right)^2 \geq 0, \tag{52}$$

which implies that the supervised fine-tuning loss $\mathcal{L}_{\text{SFT}}^t$ is logits convex at time step $t$.

## I.2 LOGITS CONVEXITY OF PG LOSS

According to Equation 26, the partial derivative of $\mathcal{L}_{\text{PG}}^t$ with respect to logit $z_\theta(a_{t,k}|s_t)$ is:

$$\frac{\partial \mathcal{L}_{\text{PG}}^t}{\partial z_{\theta,k}} = \Psi_{t,i}(\pi_{\theta,k} - \delta_{ik}), \tag{53}$$

with $z_\theta(a_{t,k}|s_t)$ simplified as $z_{\theta,k}$, and $\pi_\theta(a_{t,k}|s_t)$ simplified as $\pi_{\theta,k}$. The second derivative is:

$$\frac{\partial^2 \mathcal{L}_{\text{PG}}^t}{\partial z_{\theta,k} \partial z_{\theta,k'}} = \Psi_{t,i} \frac{\partial(\pi_{\theta,k} - \delta_{ik})}{\partial z_{\theta,k'}} = \Psi_{t,i} \pi_{\theta,k}(\delta_{kk'} - \pi_{\theta,k'}). \tag{54}$$

Notice that second derivative in Equation 54 and that in Equation 43 differ only by a scalar term $\Psi_{t,i}$. Using Equation 48, we can directly express the Hessian matrix of $\mathcal{L}_{\text{PG}}^t$ in the following form:

$$\boldsymbol{H} = \Psi_{t,i}(\text{diag}(\boldsymbol{\pi}_\theta) - \boldsymbol{\pi}_\theta \boldsymbol{\pi}_\theta^\top). \tag{55}$$

For any random vector $\mathbf{v} \in \mathbb{R}^n$, consider the quadratic form:

$$\mathbf{v}^\top \boldsymbol{H} \mathbf{v} = \Psi_{t,i} \left[ \sum_k^n \pi_{\theta,k} v_k^2 - \left( \sum_k^n \pi_{\theta,k} v_k \right)^2 \right]. \tag{56}$$

According to Equation 52:

$$\begin{cases} \mathbf{v}^\top \boldsymbol{H} \mathbf{v} \geq 0, & \text{if } \Psi_{t,i} > 0, \\ \mathbf{v}^\top \boldsymbol{H} \mathbf{v} \leq 0, & \text{if } \Psi_{t,i} < 0. \end{cases} \tag{57}$$

When $\Psi_{t,i} > 0$, $\mathcal{L}_{\text{PG}}^t$ is convex with respect to the logits. Conversely, when $\Psi_{t,i} < 0$, $\mathcal{L}_{\text{PG}}^t$ is not convex but instead concave with respect to the logits. Minimizing a concave function can lead to gradient divergence, resulting in unstable training.

### I.3 LOGITS CONVEXITY OF PG-IS LOSS

According to Equation 28, the partial derivative of $\mathcal{L}_{\text{PG-IS}}^t$ with respect to logit $z_\theta(a_{t,k}|s_t)$ is:

$$\frac{\partial \mathcal{L}_{\text{PG-IS}}^t}{\partial z_{\theta,k}} = \frac{\Psi_{t,i}}{\pi_{\theta_{\text{old}}}(a_{t,i}|s_t)} \pi_{\theta,i}(\pi_{\theta,k} - \delta_{ik}), \tag{58}$$

with $z_\theta(a_{t,k}|s_t)$ simplified as $z_{\theta,k}$, and $\pi_\theta(a_{t,k}|s_t)$ simplified as $\pi_{\theta,k}$. The second derivative is:

$$\begin{aligned}
\frac{\partial^2 \mathcal{L}_{\text{PG-IS}}^t}{\partial z_{\theta,k} \partial z_{\theta,k'}} &= \frac{\Psi_{t,i}}{\pi_{\theta_{\text{old}}}(a_{t,i}|s_t)} \frac{\partial(\pi_{\theta,i}(\pi_{\theta,k} - \delta_{ik}))}{\partial z_{\theta,k'}} \\
&= \frac{\Psi_{t,i}}{\pi_{\theta_{\text{old}}}(a_{t,i}|s_t)} [\pi_{\theta,i}(\delta_{ik'} - \pi_{\theta,k'})(\pi_{\theta,k} - \delta_{ik}) + \pi_{\theta,i}\pi_{\theta,k}(\delta_{kk'} - \pi_{\theta,k'})] \\
&= \frac{\Psi_{t,i}}{\pi_{\theta_{\text{old}}}(a_{t,i}|s_t)} \pi_{\theta,i}(-\delta_{ik}\delta_{ik'} + \pi_{\theta,k}\delta_{ik'} + \delta_{ik}\pi_{\theta,k'} - \pi_{\theta,k}\pi_{\theta,k'} + \pi_{\theta,k}(\delta_{kk'} - \pi_{\theta,k'})).
\end{aligned} \tag{59}$$

To prove the logits convexity, we need to derive the Hessian matrix $\boldsymbol{H}$ of $\mathcal{L}_{\text{PG-IS}}^t$ with respect to the logits and check if $\boldsymbol{H}$ is positive semi-definite. To construct the Hessian matrix, we need to organize the second derivatives of the loss function $\mathcal{L}_{\text{PG-IS}}^t$ with respect to the logits, which is given by:

$$\boldsymbol{H}_{k,k'} = \frac{\partial^2 \mathcal{L}_{\text{PG-IS}}^t}{\partial z_{\theta,k} \partial z_{\theta,k'}}. \tag{60}$$

From the previous derivation (Equation 59), the individual elements of the Hessian matrix $\boldsymbol{H}$ can be decomposed using elements of five matrices $\boldsymbol{A}$, $\boldsymbol{B}$, $\boldsymbol{C}$, $\boldsymbol{D}$, and $\boldsymbol{F}$:

$$\boldsymbol{H}_{k,k'} = \frac{\Psi_{t,i}}{\pi_{\theta_{\text{old}}}(a_{t,i}|s_t)} \pi_{\theta,i}(- \underbrace{\delta_{ik}\delta_{ik'}}_{\boldsymbol{A}_{k,k'}} + \underbrace{\pi_{\theta,k}\delta_{ik'}}_{\boldsymbol{B}_{k,k'}} + \underbrace{\delta_{ik}\pi_{\theta,k'}}_{\boldsymbol{C}_{k,k'}} - \underbrace{\pi_{\theta,k}\pi_{\theta,k'}}_{\boldsymbol{D}_{k,k'}} + \underbrace{\pi_{\theta,k}(\delta_{kk'} - \pi_{\theta,k'})}_{\boldsymbol{F}_{k,k'}})). \tag{61}$$

Each of the five matrices can be written in a compact form. Let

$$\boldsymbol{e}^{(i)} = [0, \ldots, 0, 1, 0, \ldots, 0]^\top \tag{62}$$

represent the standard $n$-dimension basis vector with a 1 at position $i$, and

$$\boldsymbol{\pi}_\theta = [\pi_{\theta,1}, \pi_{\theta,2}, \ldots, \pi_{\theta,n}]^\top \tag{63}$$

represent the probability distribution over the vocabulary, where $n$ is the vocabulary size. Then $\boldsymbol{A}$, $\boldsymbol{B}$, $\boldsymbol{C}$, $\boldsymbol{D}$, and $\boldsymbol{F}$ are all $n \times n$ matrices with the following structure:

$$
\begin{aligned}
\boldsymbol{A} &= \boldsymbol{e}^{(i)} \boldsymbol{e}^{(i)\top}, \\
\boldsymbol{B} &= \boldsymbol{\pi}_\theta \boldsymbol{e}^{(i)\top}, \\
\boldsymbol{C} &= \boldsymbol{e}^{(i)} \boldsymbol{\pi}_\theta^\top, \\
\boldsymbol{D} &= \boldsymbol{\pi}_\theta \boldsymbol{\pi}_\theta^\top, \\
\boldsymbol{F} &= \mathrm{diag}(\boldsymbol{\pi}_\theta) - \boldsymbol{\pi}_\theta \boldsymbol{\pi}_\theta^\top.
\end{aligned}
\tag{64}
$$

Since scaling $\boldsymbol{H}$ by any positive scalar does not affect its positive semi-definiteness, we absorb $\frac{\pi_{\theta,i}}{\pi_{\theta_{\mathrm{old}}}(a_{t,i}|s_t)}$ into $\Psi_{t,i}$ for simplicity. The Hessian matrix $\boldsymbol{H}$ has the following structure:

$$
\boldsymbol{H} = \Psi_{t,i} \left[ -\boldsymbol{e}^{(i)} \boldsymbol{e}^{(i)\top} + \boldsymbol{\pi}_\theta \boldsymbol{e}^{(i)\top} + \boldsymbol{e}^{(i)} \boldsymbol{\pi}_\theta^\top - 2\boldsymbol{\pi}_\theta \boldsymbol{\pi}_\theta^\top + \mathrm{diag}(\boldsymbol{\pi}_\theta) \right].
\tag{65}
$$

For any random vector $\mathbf{v} \in \mathbb{R}^n$, consider the quadratic form:

$$
\begin{aligned}
\mathbf{v}^\top \boldsymbol{H} \mathbf{v} &= \Psi_{t,i} \left[ -\mathbf{v}^\top \boldsymbol{e}^{(i)} \boldsymbol{e}^{(i)\top} \mathbf{v} + \mathbf{v}^\top \boldsymbol{\pi}_\theta \boldsymbol{e}^{(i)\top} \mathbf{v} + \mathbf{v}^\top \boldsymbol{e}^{(i)} \boldsymbol{\pi}_\theta^\top \mathbf{v} - 2\mathbf{v}^\top \boldsymbol{\pi}_\theta \boldsymbol{\pi}_\theta^\top \mathbf{v} + \mathbf{v}^\top \mathrm{diag}(\boldsymbol{\pi}_\theta) \mathbf{v} \right] \\
&= \Psi_{t,i} \left[ -v_i^2 + 2v_i \sum_k^n \pi_{\theta,k} v_k - 2\left( \sum_k^n \pi_{\theta,k} v_k \right)^2 + \sum_k^n \pi_{\theta,k} v_k^2 \right] \\
&= \Psi_{t,i} \left[ \underbrace{\sum_k^n \pi_{\theta,k} v_k^2 - \left( \sum_k^n \pi_{\theta,k} v_k \right)^2}_{\mathbb{D}(\mathbf{v})} - \underbrace{\left( v_i^2 - 2v_i \sum_k^n \pi_{\theta,k} v_k + \left( \sum_k^n \pi_{\theta,k} v_k \right)^2 \right)}_{(v_i - \mathbb{E}(\mathbf{v}))^2} \right] \\
&= \Psi_{t,i} \left[ \mathbb{D}(\mathbf{v}) - (v_i - \mathbb{E}(\mathbf{v}))^2 \right].
\end{aligned}
\tag{66}
$$

where $\mathbb{E}(\mathbf{v}) = \sum_k^n \pi_{\theta,k} v_k$, and $\mathbb{D}(\mathbf{v}) = \sum_k^n \pi_{\theta,k} v_k^2 - \left( \sum_k^n \pi_{\theta,k} v_k \right)^2$. According to Equation 52, we have $\mathbb{D}(\mathbf{v}) \geq 0$. Now, consider the case where $\Psi_{t,i} > 0$:

$$
\begin{cases}
\mathbf{v}^\top \boldsymbol{H} \mathbf{v} < 0, & \text{if } v_i > \mathbb{E}(\mathbf{v}) + \sqrt{\mathbb{D}(\mathbf{v})} \text{ or } v_i < \mathbb{E}(\mathbf{v}) - \sqrt{\mathbb{D}(\mathbf{v})}, \\
\mathbf{v}^\top \boldsymbol{H} \mathbf{v} \geq 0, & \text{otherwise}.
\end{cases}
\tag{67}
$$

A symmetric result holds for the case where $\Psi_{t,i} < 0$. This implies Hessian matrix $\boldsymbol{H}$ of $\mathcal{L}_{\mathrm{PG\text{-}IS}}^t$ is not positive semi-definite, which indicates that the PPO loss $\mathcal{L}_{\mathrm{PG\text{-}IS}}^t$ is not logits convex.

### I.4 LOGITS CONVEXITY OF LCO LOSS

According to Equation 30, the partial derivative of $\mathcal{L}_{\mathrm{LCO}}^t$ with respect to logit $z_\theta(a_{t,k}|s_t)$ is:

$$
\frac{\partial \mathcal{L}_{\mathrm{LCO}}^t}{\partial z_{\theta,k}} = |\Psi_{t,i}|(\pi_{\theta,k} - \pi_k'),
\tag{68}
$$

with $z_\theta(a_{t,k}|s_t)$ simplified as $z_{\theta,k}$, $\pi_\theta(a_{t,k}|s_t)$ simplified as $\pi_{\theta,k}$, and $\pi'(a_{t,k}|s_t)$ simplified as $\pi_k'$. The second derivative is as follow:

$$
\frac{\partial^2 \mathcal{L}_{\mathrm{LCO}}^t}{\partial z_{\theta,k} \partial z_{\theta,k'}} = |\Psi_{t,i}| \frac{\partial(\pi_{\theta,k} - \pi_k')}{\partial z_{\theta,k'}} = |\Psi_{t,i}| \pi_{\theta,k}(\delta_{kk'} - \pi_{\theta,k'}).
\tag{69}
$$

Notice that second derivative in Equation 69 and that in Equation 43 differ only by a scalar term $|\Psi_{t,i}|$. Using Equation 48, we can express the Hessian matrix of $\mathcal{L}_{\mathrm{LCO}}^t$ in the following form:

$$
\boldsymbol{H} = |\Psi_{t,i}|(\mathrm{diag}(\boldsymbol{\pi}_\theta) - \boldsymbol{\pi}_\theta \boldsymbol{\pi}_\theta^\top).
\tag{70}
$$

For any random vector $\mathbf{v} \in \mathbb{R}^n$, consider the quadratic form:

$$
\mathbf{v}^\top \boldsymbol{H} \mathbf{v} = |\Psi_{t,i}| \left[ \sum_k^n \pi_{\theta,k} v_k^2 - \left( \sum_k^n \pi_{\theta,k} v_k \right)^2 \right].
\tag{71}
$$

According to Equation 52:

$$\mathbf{v}^\top \boldsymbol{H} \mathbf{v} \geq 0. \tag{72}$$

It concludes that Hessian matrix $\boldsymbol{H}$ of $\mathcal{L}_{\text{LCO}}^t$ is positive semi-definite, which implies that $\mathcal{L}_{\text{LCO}}^t$ is logits convex.

## J    DERIVATION OF THE LOGIT VARIATION

We provide derivation for $\Delta z_{t,i}$ in Equation 15. First, we reformulate Equation 13 as follows:

$$\pi'(a_{t,i}|s_t) = \rho_{t,i}\pi_\theta(a_{t,i}|s_t). \tag{73}$$

Substituting this expression into Equation 14, we derive the expression for $\Delta z_{t,i}$:

$$
\begin{aligned}
\rho_{t,i}\pi_\theta(a_{t,i}|s_t) &= \frac{\exp(z_\theta(a_{t,i}|s_t) + \Delta z_{t,i})}{\sum_{k \neq i} \exp z_\theta(a_{t,k}|s_t) + \exp(z_\theta(a_{t,i}|s_t) + \Delta z_{t,i})} \\
\Rightarrow \rho_{t,i}\frac{\exp z_\theta(a_{t,i}|s_t)}{\sum_k \exp z_\theta(a_{t,k}|s_t)} &= \frac{\exp(z_\theta(a_{t,i}|s_t) + \Delta z_{t,i})}{\sum_{k \neq i} \exp z_\theta(a_{t,k}|s_t) + \exp(z_\theta(a_{t,i}|s_t) + \Delta z_{t,i})} \\
\Rightarrow \rho_{t,i}\frac{1}{\sum_k \exp z_\theta(a_{t,k}|s_t)} &= \frac{\exp \Delta z_{t,i}}{\sum_{k \neq i} \exp z_\theta(a_{t,k}|s_t) + \exp(z_\theta(a_{t,i}|s_t) + \Delta z_{t,i})} \\
\Rightarrow \exp \Delta z_{t,i} &= \rho_{t,i}\frac{\sum_{k \neq i} \exp z_\theta(a_{t,k}|s_t) + \exp(z_\theta(a_{t,i}|s_t) + \Delta z_{t,i})}{\sum_k \exp z_\theta(a_{t,k}|s_t)} \\
\Rightarrow \exp \Delta z_{t,i} &= \rho_{t,i}(1 - \pi_\theta(a_{t,i}|s_t) + \pi_\theta(a_{t,i}|s_t)\exp \Delta z_{t,i}) \\
\Rightarrow \Delta z_{t,i} &= \log \rho_{t,i} + \log \frac{1 - \pi_\theta(a_{t,i}|s_t)}{1 - \rho_{t,i}\pi_\theta(a_{t,i}|s_t)}.
\end{aligned} \tag{74}
$$

With this logit adjustment $\Delta z_{t,i}$, we can construct the target distribution $\pi'(\cdot|s_t)$ using Equation 14 and Equation 16 to control the desired update for the probability of the sampled action $a_{t,i}$.

