# OpenReview forum: "Smooth Gradients, Stable Learning: Logits Convexity for Reinforcement Learning"
_ICLR.cc/2026/Conference — ICLR 2026 Conference Withdrawn Submission_

### Official Review · Reviewer_fgdX · 2025-10-25

**Soundness:** 2
**Presentation:** 1
**Contribution:** 2
**Rating:** 2
**Confidence:** 4

**Summary:**

The paper studies RL training in LLM problems under softmax parameterization. The paper formulates the notion of logits convexity, which requires that the objective loss function is convex over the logit space of the softmax parameterization. The paper verifies this condition for SFT loss, policy gradient loss, and importance sampling loss, and argues that the logit convexity condition is essential to ensure training stability and convergence. Then, the paper introduces the logit convex optimization, and tests the method with LLM experiments.

**Strengths:**

1. The experiments are conducted thoroughly with different models and different tasks.

2. The effect of LCO is isolated, and in many cases, it is shown to improve the performance of REINFORCE, PPO, and GRPO.

3. Some analysis on the training dynamic is given, and ablation studies are presented.

**Weaknesses:**

1. The notation of this paper is mostly confusing. Please align the notation with standard RL conventions and avoid suggesting that logits are functions of the action label. Please use $a_t$ for the action selected at step $t$, instead of $a_{t,i}$, do you mean $a_{t,i}=i$? If $a_{t,i}$ is an action, which action? is it a random variable? or a constant. Please use $a_t = i$ to represent selecting action $i$ at step $t$, and $1_{a_t = i}$ for indicators. Please clarify and define variables, vectors, and functions without ambiguity.

2. All the arguments and analyses of this paper overlooked the role of expectation. For example, the paper argues that when the advantage $\Psi_{t,i}<0$, the gradient becomes larger when the policy converges. However, when the policy converges, the probability of choosing any action with a negative advantage also converges to $0$, meaning it should become very unlikely to observe a sample with negative advantage. In fact, the true policy gradient of the logits should be (eq (10) in [1]): $\frac{\partial E[L_{PG}]}{\partial z(a|s)} = \pi(a|s)(r(s,a) - \sum_{a'}\pi(a'|s)r(s,a'))$, which is $0$ for all actions when $\pi$ converges to the delta density on the optimal action. Therefore, I believe the message conveyed in the paper is not valid.

3. Some theoretical results are not well-defined or even incorrect. In definition 1, do you mean the Hessian is PSD for all $z_\theta$? In Proposition 1, the optimal logit in $R^d$ may not exist (it could be infinity, for example, in softmax parameterization), and it may not be unique, so $z_\theta^*$ is not well-defined. I feel the linear function of the logit, which is also a linear function of the parameter $\theta$, with a non-zero slope, is a counterexample to Proposition 1, i.e., the optimizing point is when the logit is infinity, but the derivative is a constant slope, which does not vanish. Also, in the proof of Proposition 1, equation (33) is not correct. It will require an additional bounded gradient condition from logit to $\theta$, which is not mentioned. In proposition (2), the inequality is not well-defined if the optimal logit is infinity.

4. Significance is missing in experiments. Please try multiple random seeds and report significance metrics.

5. The proposed approach seems to be a concatenation of a heuristic soft policy iteration with imitation learning, where both have been well-studied in the RL literature. The contribution on top of the literature is somewhat thin, and the scope is limited to mathematical reasoning tasks. It remains unclear what benefit it could have on general LLM tasks or general RL tasks. For example, could LCO improve the stability of a broader range of RL problems? Could the observation hold for non-binary or even noisy learned reward?

**Questions:**

See weaknesses

---

### Official Review · Reviewer_hcMZ · 2025-10-27

**Soundness:** 3
**Presentation:** 2
**Contribution:** 2
**Rating:** 4
**Confidence:** 3

**Summary:**

This paper studies instability in RL for LLM from the perspective of the gradient and proposes Logits Convex Optimization (LCO) that frames the target distribution to guide model updates. First, they theoretically prove that policy gradient (PG) and policy gradient with importance sampling (PG-IS) suffer from an exploded gradient norm with decreasing loss, whereas supervised fine-tuning (SFT) does not suffer. Based on the findings, they propose Logits Convex Optimization (LCO) to stabilize the gradient dynamics by guiding the model on what the target distribution is. At last, they conduct comprehensive experiments across datasets and models to illustrate the outperformance of their method

**Strengths:**

Strengths:
1.	(strong motivation) The studied question is why RL is unstable compared with SFT, which is important. Additionally, the perspective from the gradient norm is also reasonable.
2.	(clear and intuitive analysis) The analysis of the gradient dynamics is clear. They include complete mathematical inferences and vivid images for understanding. That analytical style is all through the whole paper.
3.	(comprehensive experiments) They not only evaluate their proposed method across tasks and models, but also do dynamic gradient analysis and some interesting case studies.

**Weaknesses:**

Weaknesses:
1.	(Lines 52-54) For SFT, we want the model to imitate the data because they are expert demonstrations. In comparison, we don't want the model after RL converges to the strict optimal point to preserve the generation diversity. Moreover, we have the clipping and the KL penalty to make the model change slowly in reference to the old and reference models. In that sense, we should expect different optimization behaviors for RL compared with SFT.

2.	(Section 4.2) The role of LCO loss is so different from the original KL penalty term:
a.	The original KL term reflects a global target that prevents the policy from moving too far from the reference policy. However, the LCO loss changes per batch.
b.	Following the last point, $\rho$ is more like the clipping ratio. They both define to what extent we want to update the model per batch, where \rho reflects the target and the clipping ratio reflects the boundary. If the interplay is an issue, I wonder what we should do with the clipping ratio if we already have LCO.
c.	Using PPO as an example, two forces are battling with each other: policy gradient changes the current policy, but KL stops the model from changing too much. Now with LCO, the two forces both drive the model away from the original model. Only if the model collapses to the optimum with zero advantage will there be no update.

**Questions:**

1.	(Lines 66-68) An intuitive question: if the issue comes from large gradient norms, can we resolve it by setting the max_norm hyperparameter?

2.	(Figure 2(a)) Why is there no variance reflected by the shaded area of Target Token Probability?

3.	(Section 4.2) The hyperparameter $\rho$ might be model-, question-, or stage-dependent. Is that okay to select a fixed $\rho$ based on your ablation study? Also, you should clearly state that $\rho$ is a hyperparameter and how you select it.

4.	(Section 6.2) What are the definitions of Entropy and Action Probabilities? Why is that entropy = 0 and sampled action probability = 1 before the training?

5.	(Figure 7)
a.	Are the drops of PG and PG-IS because of overfitting? They always reach the highest before a significant drop. Is early stopping enough to fix the issue?
b.	How do you understand several vibrations of the green line of LCO?

6.	(Figure 8) Unlike Figure 6(a), where the metrics are correlated with each other, why is it that the correlation disappears? For example, clear variations in the gradient norm, entropy & action probability, and Pass@1 happen together.

7.	(Lines 68-69 and Section 6.4) Some work proposes to focus more on the tokens with high uncertainty (e.g., Beyond the 80/20 Rule: High-Entropy Minority Tokens Drive Effective Reinforcement Learning for LLM Reasoning). How do you interpret the difference between your propositions?

---

### Official Review · Reviewer_yf1n · 2025-11-01

**Soundness:** 2
**Presentation:** 3
**Contribution:** 2
**Rating:** 2
**Confidence:** 4

**Summary:**

This paper investigates the instability commonly observed when training large language models (LLMs) with reinforcement learning (RL), as compared to supervised fine-tuning (SFT). The authors identify a property they term logits convexity, which they argue is present in SFT but absent in standard RL objectives. To address this gap, they propose a new regularization technique called Logits Convex Optimization (LCO), designed to encourage convexity in the logits space and thereby stabilize RL training.

**Strengths:**

- The paper tackles a highly relevant problem: the instability of RL fine-tuning for LLMs. Framing this issue through gradient dynamics and comparing it directly to SFT is a valuable and underexplored direction.
- Despite limitations in the theoretical analysis, the empirical observations (e.g., differing gradient norm behaviors between SFT and RL) may inspire future work on stabilizing policy gradient methods.
- The proposed LCO regularizer is simple to implement and shows consistent (if modest) improvements over vanilla REINFORCE.

**Weaknesses:**

- Inaccurate policy gradient formulation: Equations (3) and (4) do not correspond to standard policy gradient losses. Notably, the expectation over trajectories is missing, which is critical for correct gradient computation. This omission affects the validity of the subsequent gradient norm analysis in Section 3. A more rigorous treatment can be found in Mei et al. (ICML 2020) [1], which the authors should reference.

- As a KL regularization term, it is not surprising to see that LCO can stablize training by reducing gradient variance/norm of REINFORCE, thus improves the performance of REINFORCE. Howevre, for more advanced algorithm (e.g., GRPO and PPO) which already incoprated many techniques to stabilize training, LCO is less effective and provides no significant performance improvement.

- Flawed theoretical reasoning in Section 3:
    - The claim that the sign of the advantage estimate $\psi_{t,i}$ drives instability is misleading. Since the gradient norm depends on $|\psi_{t,i}|$, it is invariant to the sign of $\psi_{t,i}$. Thus, the sign alone cannot explain the observed differences in gradient dynamics.
    - In Section 3.2, the paper attributes the non-monotonic gradient norm behavior to negative advantages. However, this behavior can also arise with positive advantages if the initial policy assigns low probability to the sampled action (i.e., $π(a_t|s_t) < 0.5$). In fact, post-pretraining policies typically assign high probabilities to plausible actions (as shown in Figures 3a and 4a–b). The observed differences may therefore stem more from initialization and policy entropy than the sign of $\psi_{t,i}$.

- Limited empirical impact on modern RL algorithms: While LCO improves REINFORCE, its benefits diminish when applied to more advanced methods like GRPO or PPO, which already incorporate variance reduction, trust regions, or KL penalties. The paper does not convincingly demonstrate that LCO provides meaningful gains beyond what existing stabilizing mechanisms already achieve.

1. On the Global Convergence Rates of Softmax Policy Gradient Methods. Jincheng Mei, Chenjun Xiao, Csaba Szepesvari, Dale Schuurmans. ICML, 2020.

**Questions:**

- In the paper, $\psi_{t,i}$ is used to denote either the return or the advantage. Consider a setting with non-negative rewards (e.g., $r \in [0,1]$) and no advantage normalization under such conditions, $\psi_{t,i}$ would be non-negative throughout training. According to the paper's theoretical framework (Section 3), does this imply that RL training should be inherently stable in such environments? If not, what other factors might dominate instability, and how does LCO address them?

- Equation (17) defines the LCO regularizer using a weighted $\mathrm{KL}(\pi' \| \pi)$, where $\pi'$ is the target policy and $\pi$ is the current policy. Would using the reverse KL, $\mathrm{KL}(\pi \| \pi')$, alter the theoretical properties (e.g., convexity guarantees) or empirical performance? Could the authors comment on this design choice and whether alternatives were explored?

- In Tables 1 and 2, several results show that applying LCO increases pass@1 while decreasing pass@8. Why would this happen? Additionally, to assess result reliability, could you provide confidence intervals or error bars (e.g., across random seeds) for these metrics?

- Section 6.2 focuses on REINFORCE. How about the results for PPO or GRPO?

---

### Official Review · Reviewer_3gTJ · 2025-11-03

**Soundness:** 1
**Presentation:** 2
**Contribution:** 1
**Rating:** 2
**Confidence:** 4

**Summary:**

The paper says a property called logits convexity causes the stability gap between SFT and RL. This is the local convexity of the loss in the logits. The paper argues that standard policy-gradient objectives (REINFORCE/PPO/GRPO) lack this property. It proposes Logits Convex Optimization (LCO). This method builds a target distribution each step. It changes only the sampled action's logit using a ratio $ \rho_{t,i} $. Then it minimizes $ \mathrm{KL}(\pi' \,\|\, \pi_\theta) $ scaled by $|\Psi_{t,i}|$. The paper reports smoother gradients. It also shows modest gains on math-reasoning and a few OOD tasks.

**Strengths:**

* The paper gives a clear derivation of the SFT Hessian $H=\mathrm{diag}(\pi)-\pi\pi^\top$. It also derives its PSD property. This explains shrinking gradients under SFT. The exposition is standard and technically correct.
* LCO’s loss is a KL projection toward a constructed target. The paper shows its Hessian has the same PSD form, up to a positive scalar. This gives logits-convexity and smooth gradient decay.

**Weaknesses:**

1.  The central observation is textbook. Softmax cross-entropy is convex in the logits. It yields vanishing logits gradients near optimum. Rebranding this as a new principle adds little value.

2.  The objective minimizes $ \mathrm{KL}(\pi' \,\|\, \pi_\theta) $ to a target distribution. This is standard in trust-region or mirror-descent policy search. The paper frames this as novel. It does not engage that literature. The method's uniqueness rests only on how $\pi'$ is engineered.

3.  The curvature analysis targets a PG-IS surrogate. It proves it is not logits-convex. But PPO/GRPO use clipping. Clipping changes curvature and non-smoothness properties. The paper still generalizes conclusions to “widely used algorithms such as PPO/GRPO.” The paper acknowledges that clipping is applied in experiments. This is a category error.

4.  $\pi'$ is built by only shifting the sampled action’s logit. It uses $\Delta z_{t,i}$ computed from a hand-picked ratio $\rho_{t,i}$. Non-sampled actions are just re-normalized. There is no derivation from an optimality principle, like exponentiated-advantage. Ignoring competing actions’ structure risks brittle behavior in long-horizon generation.

5.  The algorithm discretizes the advantage sign. It uses global constants chosen by a small-grid search (e.g., $ \rho^+=1.8, \rho^-=0.9 $). The magnitude only influences a scalar prefactor $|\Psi_{t,i}|$ outside the KL. This invites overfitting to an eval set. It also weakens credit assignment.

**Questions:**

1.  Can you derive LCO as an optimal KL-regularized policy-improvement step? Spell out the gap from exponentiated-advantage targets. Currently, $\pi'$ is engineered by a one-logit adjustment.
2.  Analyze the clipped PPO/GRPO objectives directly. Do your curvature claims hold once clipping is modeled? If not, what part of the instability comes from clipping versus concavity?
3.  Why throw away advantage magnitude in $\rho$? Could choices like $\rho=\exp(\beta\,\Psi_{t,i})$ or a learned schedule remove the need for manual grid search? Provide ablations.
4.  Clarify the dataset splits used to tune $\rho$. Did any test set metric inform hyperparameters? An example is monitoring the MATH500 test during training. Re-report with a clean validation split.
5.  Report all main results over 3 or more seeds with CIs. Show per-seed trajectories for gradient norms and entropy.

---

### Note · Authors · 2025-12-17

I have read and agree with the venue's withdrawal policy on behalf of myself and my co-authors.